# Different pathways for engulfment and endocytosis of liquid droplets by nanovesicles

Rikhia Ghosh [1,2], Vahid Satarifard[1,3] & Reinhard Lipowsky [1]✉

During endocytosis of nanoparticles by cells, the cellular membranes engulf the particles, thereby forming a closed membrane neck that subsequently undergoes fission. For solid nanoparticles, these endocytic processes have been studied in some detail. Recently, such processes have also been found for liquid and condensate droplets, both in vitro and in vivo. These processes start with the spreading of the droplet onto the membrane followed by partial or complete engulfment of the droplet. Here, we use molecular dynamics simulations to study these processes at the nanoscale, for nano-sized droplets and vesicles. For both partial and complete engulfment, we observe two different endocytic pathways. Complete engulfment leads to a closed membrane neck which may be formed in a circular or strongly non-circular manner. A closed circular neck undergoes fission, thereby generating two nested daughter vesicles whereas a non-circular neck hinders the fission process. Likewise, partial engulfment of larger droplets leads to open membrane necks which can again have a circular or non-circular shape. Two key parameters identified here for these endocytic pathways are the transbilayer stress asymmetry of the vesicle membrane and the positive or negative line tension of the membrane-droplet contact line.

The cells of higher organisms contain a large variety of intracellular bodies or organelles. Conventional organelles are enclosed by membranes that partition the interior of the cell into distinct aqueous compartments. Recently, biomolecular condensates have been discovered that represent membraneless organelles and behave like liquid droplets. Examples for these kinds of condensates include germ P-bodies[1], nucleoli[2], and stress granules[3], as reviewed in ref. [4]. These biomolecular condensates are believed to form via liquid–liquid phase separation in the cytoplasm and can be reconstituted in vitro[5–8]. They are enriched in certain types of proteins that have intrinsically disordered domains and interact via multivalent macromolecular interactions[4,7–10].

The formation of biomolecular condensates in cells resembles the formation of molecular condensates in aqueous two-phase systems, also called aqueous biphasic systems, that have been used for a long time in biochemical analysis and biotechnology[11] and are intimately related to water-in-water emulsions[12]. Aqueous two-phase systems based on biopolymers such as PEG and dextran undergo phase separation when the weight fractions of the polymers exceed a few percent. The corresponding interfacial tension varies over several orders of magnitude, reflecting the vicinity of a critical demixing point in the phase diagram[13–16]. Indeed, the interfacial tension vanishes at the critical point, attains ultralow values of the order of $10^{-4}$ mN/m close to this point, and continuously increases up to several mN/m as we move further away from the critical point[14].

Liquid droplets lead to the wetting and molding of biomembranes as first observed for molecular condensates arising from aqueous phase separation within giant unilamellar vesicles (GUVs)[17–21]. GUVs have a diameter of many micrometers and their morphological responses can be directly observed in the optical microscope.

[1]Max Planck Institute of Colloids and Interfaces, Science Park Golm, 14424 Potsdam, Germany. [2]Icahn School of Medicine Mount Sinai, 1 Gustave L. Levy Pl, New York, NY 10029, USA. [3]Yale Institute for Network Science, Yale University, New Haven, CT 06520, USA. ✉e-mail: lipowsky@mpikg.mpg.de

Furthermore, these shape transformations can be understood in a quantitative manner using the theory of curvature elasticity[18,22,23]. In the context of biomolecular condensates, evidence for membrane remodeling and endocytosis by condensate-membrane interactions has been provided for P-bodies that adhere to the outer nuclear membrane[1], for lipid vesicles within a synapsin-rich liquid phase[24], for TIS granules interacting with the endoplasmic reticulum[25], and for condensates at the plasma membrane[26–28]. All of these experimental studies looked at membrane remodeling processes on the scale of many micrometers. Quite recently, super-resolution microscopy has been used to study tubular protrusions of giant vesicles[29,30]. These tubules had a width between 40 and 150 nm.

In the present study, we use coarse-grained molecular dynamics, so-called Dissipative Particle Dynamics (DPD)[31,32], in order to investigate the morphological responses of nanovesicles when they come into contact with small droplets. These droplets arise via liquid–liquid phase separation in a binary liquid mixture, for which the phase diagram has been determined in a previous study[33]. The nanovesicles are taken to have a constant size of 37 nm, which is determined by the total number of lipids used to assemble the bilayer membranes, whereas the diameter of the droplets is controlled by the chosen solute concentration. As explained in the "Methods" section, we generate three different droplet diameters that vary between 11 nm and 19 nm by using three different solute concentrations. Initially, both the nanovesicle and the nanodroplet have a spherical shape as displayed in Fig. 1a. When the droplet comes into close contact with the vesicle, it adheres to the vesicle and spreads onto the vesicle membrane, see Fig. 1b. After the vesicle–droplet couple has reached its new equilibrium state as in Fig. 1c, the contact line between the adhering droplet and the vesicle divides the vesicle membrane up into two segments, (i) the $\alpha\gamma$ segment, which represents the contact area between the $\alpha$ droplet and the vesicle as well as (ii) the $\beta\gamma$ segment between the vesicle and the aqueous bulk phase $\beta$. The lateral extension of the contact area depends on the contact angle between the droplet and the membrane: the contact area becomes large for small contact angles and small for large angles. A large contact area corresponding to a low contact angle has been observed for several types of condensate droplets[34,35]. In general, the contact angle can attain a value between 0° for complete wetting by the $\alpha$ droplet and 180° for complete dewetting of this droplet. In the systems studied here, the contact angle is close to 90° as can already be concluded from Fig. 1c. The precise numerical values of the contact angles as measured in the simulations confirm this conclusion.

The engulfment process represents an important step towards droplet endocytosis. During this process, both the droplet and the vesicle membrane become deformed into nonspherical shapes. Based on the resulting vesicle–droplet morphology, we identify different pathways of endocytosis. Sufficiently small droplets are completely engulfed by the membrane which then forms a closed membrane neck, following two different pathways. One pathway is provided by axisymmetric shapes of vesicle and droplet as well as subsequent fission of the circular membrane neck which divides the nanovesicle into two nested daughter vesicles. Another pathway leads to nonaxisymmetric shapes and noncircular membrane necks which hinder the fission process. Larger droplets are only partially engulfed by the membrane which now forms an open membrane neck, following again two pathways with circular and noncircular necks. A detailed analysis reveals that these pathways depend primarily on two key parameters, on the transbilayer stress asymmetry of the vesicle membrane and on the line tension of the contact line between membrane and droplet.

Because of their small size, the nanovesicles studied here have a high membrane curvature that applies to both synthetic liposomes, so-called small unilamellar vesicles (SUVs), and to extracellular vesicles as provided by exosomes and microvesicles. SUVs can be produced by a variety of preparation methods[36,37]. Exosomes and extracellular vesicles are released from almost every type of living cell[38] and are crucial for the chemical communication between cells. These vesicles have been intensely studied as possible biomarkers for diseases[39,40] and as potential drug delivery systems[41,42]. Our simulation study demonstrates that the endocytosis of liquid droplets is possible both for small liposomes and for extracellular vesicles.

A variety of computational approaches has been previously used to study the engulfment of solid or rigid nanoparticles by lipid bilayers and nanovesicles. Engulfment of such particles has been investigated by Brownian dynamics[43], DPD simulations[44–46] and Monte Carlo simulations[47]. DPD simulations have also been applied to the translocation of relatively small nanoparticles through lipid bilayers[48]. The main difference between liquid droplets and solid or rigid nanoparticles is that the latter particles have a finite shear modulus which generates local restoring forces against elastic deformations. Furthermore, none of these previous studies on rigid nanoparticles addressed the two key parameters, transbilayer stress asymmetry and contact line tension, that we identify here for the engulfment and endocytosis of liquid droplets by nanovesicles.

Our paper is organized as follows. We first address the assembly of nanovesicles with $N_{\text{ol}}$ lipids in the outer leaflet and $N_{\text{il}}$ lipids in the inner leaflet for fixed total lipid number $N_{\text{ol}} + N_{\text{il}}$. We focus on four different values of $N_{\text{ol}}$ which correspond to four different transbilayer stress asymmetries between the two leaflet tensions. We then add a liquid droplet as in Fig. 1a and let it spread onto the membrane of each

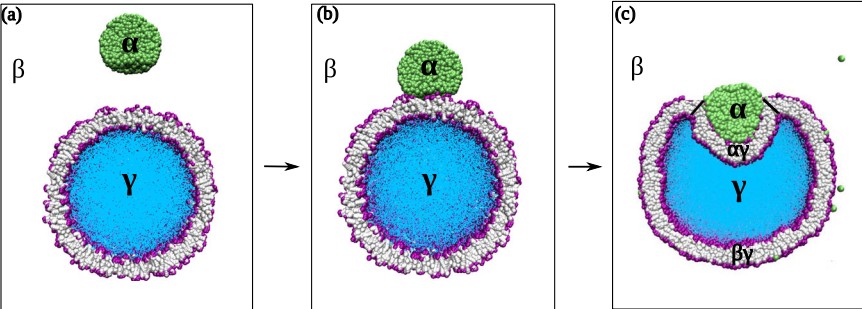

**Fig. 1 | Nanovesicle formed by a lipid bilayer with purple headgroups and gray hydrocarbon chains.** The vesicle encloses the aqueous solution $\gamma$ (blue) and interacts with the liquid nanodroplet $\alpha$ (green). Both the nanodroplet and the nanovesicle are immersed in the aqueous bulk phase $\beta$ (white): **a** Initially, the droplet is well separated from the vesicle which implies that the outer leaflet of the bilayer is only in contact with the $\beta$ phase. In this situation, both the nanodroplet and the nanovesicle have a spherical shape; **b** When the droplet is attracted towards the vesicle, it spreads onto the lipid bilayer, thereby forming an increasing contact area with the vesicle membrane; and **c** Partial engulfment of the droplet by the membrane after the vesicle–droplet couple has relaxed to a new stable state. The contact area between bilayer and $\alpha$ droplet defines the $\alpha\gamma$ segment of the bilayer membrane whereas the rest of the bilayer represents the $\beta\gamma$ segment still in contact with the $\beta$ phase. This engulfment process deforms both the nanovesicle and the nanodroplet into nonspherical shapes.

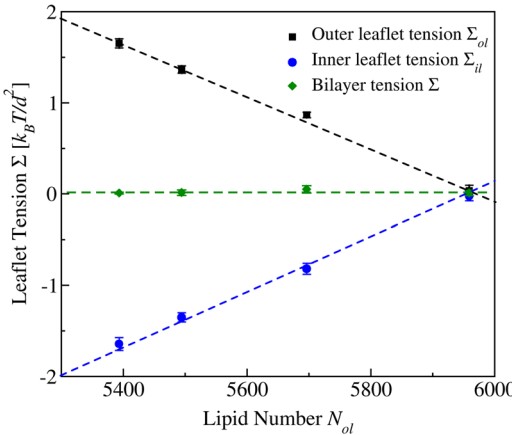

**Fig. 2 | Stress asymmetry of tensionless bilayers for spherical nanovesicles with volume $v = v_0$:** leaflet tensions $\Sigma_{ol}$ (black squares) and $\Sigma_{il}$ (blue circles) of the outer and inner leaflet as a function of the lipid number $N_{ol}$ assembled in the outer leaflet which implies the lipid number $N_{il} = 10,100 - N_{ol}$ in the inner leaflet. For $5400 \leq N_{ol} < 5963$, the outer leaflet is stretched by the positive tension $\Sigma_{ol}$ whereas the inner leaflet is compressed by the negative tension $\Sigma_{il}$. Both leaflet tensions vanish for $N_{ol} = 5963$. In all cases, the bilayer tension $\Sigma = \Sigma_{ol} + \Sigma_{il}$ (green diamonds) is close to zero. The numerical values for the volume $v_0$ as well as for the leaflet and bilayer tensions are provided in Table S1. Each data point represents the mean value over $n = 15$ statistically independent samples obtained from three replicated simulations. Each error bar is the standard error of the mean (SEM).

nanovesicle. Using the general isoperimetric inequality from differential geometry, we show that complete engulfment of the droplet is only possible for sufficiently small droplets which do not exceed a certain maximal droplet volume. The engulfment of these small droplets are then characterized by two different endocytic pathways. Next, we vary the size of the droplets and show that sufficiently large droplets are only partially engulfed but exhibit again two different endocytic pathways. Furthermore, we determine the bilayer tensions of the two membrane segments as well as the line tension of the contact line between droplet and vesicle membrane. At the end, we summarize our results and discuss how these results can be generalized to more complex vesicle–droplet systems.

## Results

### Nanovesicles with transbilayer asymmetry

Our simulation protocol starts with the assembly of lipid bilayers that form spherical nanovesicles with a diameter of $46.6\,d$ or 37 nm, where the bead diameter $d = 0.8$ nm represents the basic length scale in the simulations, see "Methods" section for more details on the simulation parameters. Each bilayer consists of two leaflets, with $N_{ol}$ lipids in the outer leaflet and $N_{il}$ lipids in the inner leaflet. We then vary the lipid numbers $N_{ol}$ and $N_{il}$, keeping the total lipid number constant with $N_{ol} + N_{il} = 10\,100$. Therefore, the lipid number $N_{ol}$ in the outer leaflet will be taken as the basic control parameter for the assembly of the nanovesicles. The number $N_{ol}$ is increased and decreased by reshuffling lipids from the inner to the outer leaflet and vice versa. The volume of the vesicle is measured in terms of the rescaled and dimensionless volume parameter $v$ as defined by Eq. (2) in the "Methods" section. Using this definition, the initial spherical vesicle has the volume $v = 1$.

In order to avoid bilayer rupture, we consider tensionless bilayers that experience a mechanical bilayer tension close to zero. To determine this bilayer tension, we compute the stress profile across the bilayer and integrate this stress profile over the radial coordinate $r$ that measures the distance from the center of the spherical nanovesicle, see Eq. (4) in the "Methods" section. By examining these stress profiles and the resulting bilayer tensions for different vesicle volumes $v$, we

identify the specific volume, $v = v_0$, for which the bilayer tension, $\Sigma$, becomes close to zero. This procedure leads to the values of $v_0$ and $\Sigma$ as displayed in Table S1 of the Supplementary Information.

It is important to realize that the two leaflets of a tensionless bilayer will typically experience significant leaflet tensions, $\Sigma_{ol}$ and $\Sigma_{il}$. Indeed, because the bilayer tension, $\Sigma$, is equal to the sum of the two leaflet tensions, a tensionless bilayer with $\Sigma = 0$ implies that $\Sigma_{il} = -\Sigma_{ol}$ and that one leaflet tension is positive whereas the other leaflet tension is negative, corresponding to one stretched and one compressed leaflet, respectively. In the following, we will focus on nanovesicles for which the outer leaflet is stretched by a positive outer leaflet tension $\Sigma_{ol} > 0$ whereas the inner leaflet is compressed by a negative inner leaflet tension $\Sigma_{il} < 0$, see Fig. 2 and Table S1.

Inspection of Fig. 2 shows that the lipid numbers $N_{ol}$ and $N_{il} = 10,100 - N_{ol}$ used to assemble the outer and inner leaflets determine the leaflet tensions $\Sigma_{ol}$ and $\Sigma_{il}$ and thus the stress asymmetry between the two leaflets. However, the magnitude of this asymmetry is difficult to estimate directly from these numbers because the area of the outer leaflet is larger than the one of the inner leaflet which implies that the outer leaflet can always accommodate more lipids than the inner one. To avoid this geometric effect, we define the symmetric bilayer of the nanovesicle to consist of two tensionless leaflets with both $\Sigma_{ol}$ and $\Sigma_{il}$ close to zero[49], corresponding to the rightmost data point in Fig. 2. As shown in this figure, both leaflet tensions vanish simultaneously for lipid numbers $N_{ol} = 5963$ and $N_{il} = 10,100 - 5963 = 4137$ in the outer and inner leaflets. Nanovesicles formed by a tensionless bilayer with a stretched outer leaflet and a compressed inner one are then obtained for $N_{ol} < 5963$ and $N_{il} > 4137$. In the following, we will focus on nanovesicles with outer lipid numbers $N_{ol} = 5400$, 5500, and 5700, in addition to the symmetric reference state with $N_{ol} = 5963$, corresponding to the four sets of data in Fig. 2.

### Droplet endocytosis by nanovesicles

We now consider one of the spherical nanovesicles with initial volume $v = 1$ and expose this vesicle to an exterior solution that consists of a binary mixture of water and solutes. The mixture undergoes liquid–liquid phase separation into a solute-rich phase $\alpha$ which coexists with the water-rich phase $\beta$[33]. For relatively small solute concentrations, the $\alpha$ phase represents the minority phase which forms a small droplet within the bulk phase $\beta$ as in Fig. 1a. The droplet is bounded by an $\alpha\beta$ interface with interfacial tension $\Sigma_{\alpha\beta}$. On the molecular scale, this interface is not smooth but has an intrinsic width or fuzziness which is of the order of $\sqrt{k_B T / \Sigma_{\alpha\beta}}$ with the thermal energy $k_B T$[50,51]. Furthermore, the size of the $\alpha$ droplet can be directly controlled by the solute concentration as explained in the "Methods" section. We study three different solute concentrations for which the $\alpha$ droplets attain the three diameters $14d$ or 11.2 nm, $18.7d$ or 15 nm, and $24.5d$ or 19.6 nm. To ensure that all droplets have a diameter that is large compared to the intrinsic interfacial width, we choose a tie line in the phase diagram for which the interfacial tension $\Sigma_{\alpha\beta}$ has a relatively large value of about $3k_B T/d^2$ or 19 mN/m at room temperature, see Table 1 below.

After the spherical vesicle has been slightly deflated to volume $v = v_0$ and the vesicle membrane experiences almost zero bilayer tension, the vesicle and the $\alpha$ droplet are brought into contact. The droplet then starts to spread on the vesicle membrane, see Fig. 1b, and to become engulfed by the membrane. After 10 μs, the vesicle–droplet couple reaches a new equilibrium state as illustrated in Fig. 1c. One should note that this state is axisymmetric and corresponds to partial engulfment of the droplet, which still forms an extended $\alpha\beta$ interface with the aqueous bulk phase $\beta$. This $\alpha\beta$ interface continuously shrinks when the vesicle volume is further reduced and the $\alpha$ droplet becomes more and more engulfed by the vesicle membrane. For a sufficiently small droplet or a sufficiently large vesicle membrane, this process

**Table 1 | Numerical values of surface and line tensions corresponding to the data displayed in Figs. 6 and 7: Diameter $D_{dr}$ of droplets; lipid numbers $N_{ol}$ and $N_{il}$ in the outer and inner leaflets, encoding the transbilayer asymmetry of the nanovesicle as in Fig. 2; interfacial tension $\Sigma_{\alpha\beta}$ of liquid–liquid interface between $\alpha$ droplet and aqueous bulk phase $\beta$; bilayer tensions $\Sigma_{\alpha\gamma}$ and $\Sigma_{\beta\gamma}$ of the two membrane segments $\alpha\gamma$ and $\beta\gamma$ defined in Fig. 1; as well as contact line tension $\lambda$**

| $D_{dr}$ | $N_{ol}$ | $N_{il}$ | $\Sigma_{\alpha\beta}$ | $\Sigma_{\alpha\gamma}$ | $\Sigma_{\beta\gamma}$ | $\lambda$ | $N_{ol}^{=}$ | $N_{ol}^{[0]}$ |
|---|---|---|---|---|---|---|---|---|
| 11.2 nm | 5400 | 4700 | 2.82 ± 0.04 | 0.66 ± 0.05 | 1.32 ± 0.07 | 19.44 ± 4.20 | 5600 | 5582 |
|  | 5500 | 4600 | 2.84 ± 0.02 | 0.86 ± 0.10 | 1.11 ± 0.04 | 7.04 ± 0.31 |  |  |
|  | 5700 | 4400 | 2.85 ± 0.01 | 1.15 ± 0.11 | 0.86 ± 0.12 | −10.03 ± 2.09 |  |  |
|  | 5963 | 4137 | 2.85 ± 0.04 | 1.75 ± 0.05 | 0.57 ± 0.06 | −23.25 ± 5.1 |  |  |
| 15 nm | 5400 | 4700 | 2.94 ± 0.01 | 0.97 ± 0.04 | 1.47 ± 0.01 | 17.49 ± 5.2 | 5560 | 5572 |
|  | 5500 | 4600 | 2.96 ± 0.04 | 1.17 ± 0.08 | 1.33 ± 0.09 | 6.17 ± 3.8 |  |  |
|  | 5700 | 4400 | 2.93 ± 0.03 | 1.47 ± 0.02 | 1.06 ± 0.08 | −11.35 ± 4.11 |  |  |
|  | 5963 | 4137 | 2.96 ± 0.04 | 1.92 ± 0.05 | 0.76 ± 0.06 | −26.28 ± 5.1 |  |  |
| 19.6 nm | 5400 | 4700 | 2.9 ± 0.1 | 1.14 ± 0.03 | 1.39 ± 0.08 | 15.2 ± 4.1 | 5500 | 5538 |
|  | 5500 | 4600 | 2.95 ± 0.06 | 1.33 ± 0.03 | 1.32 ± 0.10 | 3.26 ± 2.4 |  |  |
|  | 5700 | 4400 | 2.96 ± 0.04 | 1.72 ± 0.08 | 1.08 ± 0.06 | −13.61 ± 4.9 |  |  |
|  | 5963 | 4137 | 2.93 ± 0.01 | 2.02 ± 0.08 | 0.88 ± 0.05 | −26.25 ± 5.8 |  |  |

The three surface tensions are given in units of $k_B T/d^2$, the line tension in units of $k_B T/d$. For lipid number $N_{ol} = N_{ol}^{=}$ in the outer leaflet, the two membrane segments experience the same segment tensions, see vertical broken lines in Fig. 6. For $N_{ol} = N_{ol}^{[0]}$, the line tension vanishes, see vertical broken lines in Fig. 7. The lipid number $N_{ol}^{[0]}$ is equal to the lipid number $N_{ol}^{[=]}$ within the accuracy of our computational approach.

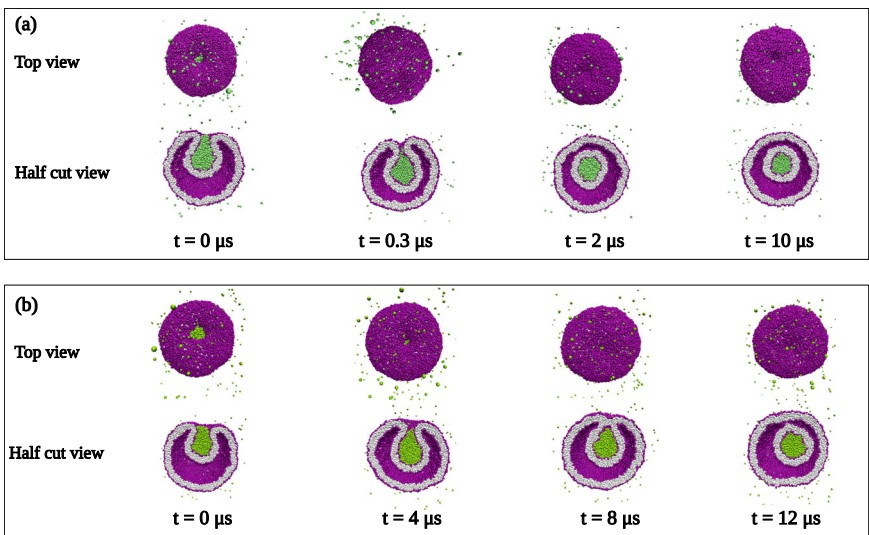

**Fig. 3 | Complete axisymmetric engulfment of nanodroplets (green) followed by division of nanovesicle (purple-gray) into two nested daughter vesicles.** **a** Nanovesicle formed by a lipid bilayer with $N_{ol}$ = 5400 lipids in the outer and $N_{il}$ = 4700 lipids in the inner leaflet. At time $t = 0$, the droplet is partially engulfed by the vesicle membrane, which forms an open membrane neck. At $t = 0.3$, the neck closes and the droplet becomes completely engulfed. The neck then undergoes fission at $t = 2\,\mu s$, thereby generating a small intraluminal vesicle around the droplet. More details on the shape evolution are provided by the Movies S1 and S2; and **b** Bilayer with $N_{ol}$ = 5500 lipids in the outer and $N_{il}$ = 4600 lipids in the inner leaflet. The membrane neck now closes at $t = 4\,\mu s$ and undergoes fission at $t = 9\,\mu s$, again generating a small intraluminal vesicle around the droplet. More details on this shape evolution are provided by the Movies S3 and S4, which also demonstrates that the morphology with two nested vesicles remains unchanged up to 30 μs. In both (**a**) and (**b**), the nanovesicles have a size of 37 nm, the droplets have a diameter of 11.2 nm, and the vesicle volume is equal to $v = 0.6$ during the whole endocytic process.

leads to complete engulfment of the droplet, with the $\alpha\beta$ interface being replaced by a closed membrane neck.

On the other hand, if the droplet is too large or the vesicle membrane too small, the engulfment process cannot be completed for purely geometric reasons. Indeed, in order to achieve complete engulfment, the droplet volume must not exceed a certain value that depends on the surface areas of the two membrane segments, as shown in the "Methods" section, using the isoperimetric inequality[52,53] from differential geometry, which applies to general surface shapes in three-dimensional space. When complete engulfment is not possible, the engulfment process is arrested in a partially engulfed state with an open membrane neck. Both for complete and for partial engulfment,

we observe two different pathways depending on the transbilayer asymmetry encoded in the lipid numbers $N_{ol}$ and $N_{il}$ or, equivalently, in the leaflet tensions $\Sigma_{ol}$ and $\Sigma_{il}$. For complete engulfment, these two pathways are displayed in Figs. 3 and 4. The first pathway in Fig. 3 proceeds via axisymmetric shapes of the vesicle–droplet couple, which leads to circular contact lines and circular membrane necks that subsequently undergo membrane fission. The second pathway in Fig. 4 leads to nonaxisymmetric shapes with noncircular contact lines and membrane necks which act to prevent the fission process.

The shape evolution of the vesicle–droplet couple also depends on the size of the $\alpha$ droplet. The shapes in Figs. 3 and 4 were obtained for small nanodroplets with diameter $14d$ or 11.2 nm. Larger droplets

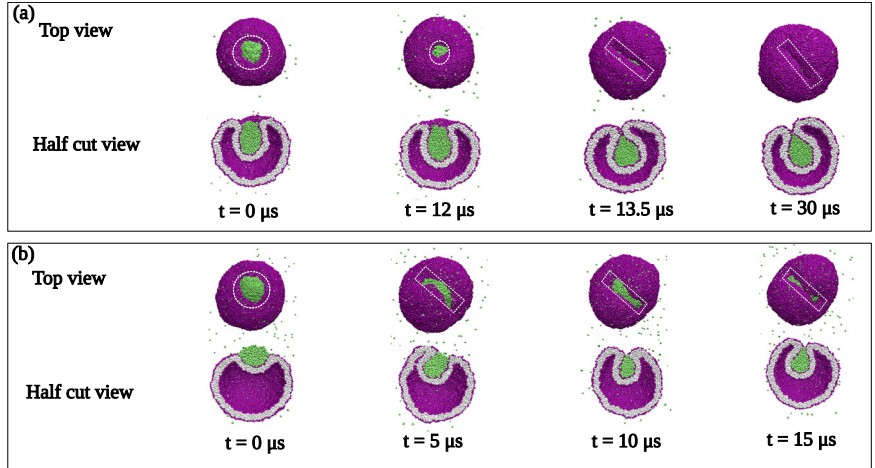

**Fig. 4 | Complete nonaxisymmetric engulfment of nanodroplets (green) without division of nanovesicle (purple–gray). a** Nanovesicle formed by a lipid bilayer with $N_{ol}$ = 5700 lipids in the outer and $N_{il}$ = 4400 lipids in the inner leaflet. At time $t$ = 0, we start with vesicle volume $v$ = 0.7, for which we observe an axisymmetric morphology of vesicle and partially engulfed droplet (green) that persists until $t$ = 12 μs, see white dashed circles around the contact lines. At $t$ = 12 μs, we reduce the vesicle volume from $v$ = 0.7 to $v$ = 0.6, which leads to complete engulfment of the droplet and to a nonaxisymmetric morphology of the vesicle–droplet couple. The broken rotational symmetry is directly visible from the strongly noncircular and highly elongated contact line, see white dashed rectangles around the contact lines at $t$ = 13.5 μs and $t$ = 30 μs. More details on the shape evolution are provided by the Movies S5 and S6; and **b** Nanovesicle formed by $N_{ol}$ = 5963 lipids in the outer and $N_{il}$ = 4137 lipids in the inner leaflet, with the vesicle volume being kept at the constant value $v$ = 0.7. At $t$ = 0, the droplet is partially engulfed by the vesicle membrane with an axisymmetric contact line, see white dashed circle. The axial symmetry is broken at $t$ = 5 μs, as follows from the strongly noncircular and highly elongated contact lines for $t \geq 5$ μs, see white dashed rectangles. More details on the shape evolution are provided by the Movies S7 and S8. In both (**a**) and (**b**), the nanovesicle has a size of 37 nm and the nanodroplet has a diameter of 11.2 nm.

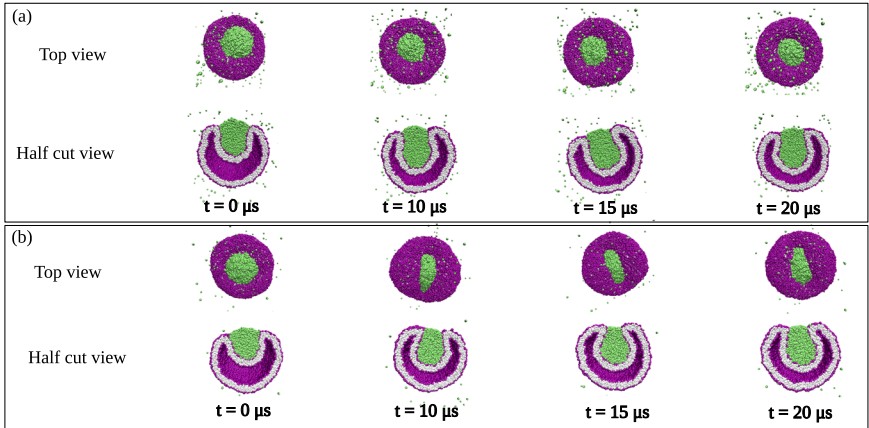

**Fig. 5 | Partial engulfment of large nanodroplets (green) with a diameter of 19.6 nm by vesicle membranes (purple-gray) with different transbilayer asymmetries as controlled by the outer lipid number $N_{ol}$.** The large size of the droplet prevents complete engulfment: **a** Nanovesicle with $N_{ol}$ = 5400 and $N_{il}$ = 4700 as well as volume $v$ = 0.45 at different time points $t$. In this case, the vesicle–droplet couple maintains its axisymmetric geometry and exhibits a circular contact line during the whole partial engulfment process. The same behavior is observed for larger volumes $v$ > 0.45; and **b** Nanovesicle with the same volume $v$ = 0.45 but a different transbilayer asymmetry corresponding to $N_{ol}$ = 5700 and $N_{il}$ = 4400 at different time points $t$. Thus, compared to (**a**) only 300 lipids have been reshuffled from the inner to the outer leaflet but the rotational symmetry is now broken leading to a noncircular contact line and to an open but noncircular membrane neck for the partially engulfed state.

with a diameter of 19.6 nm are displayed in Fig. 5. In this case, the droplets are only partially engulfed by the membrane, which now forms an open membrane neck, following again two different pathways with circular and noncircular contact lines.

**Axisymmetric engulfment and fission of membrane neck**
First, we study vesicle–droplet systems with $N_{ol}$ = 5400 and $N_{ol}$ = 5500 lipids in their outer leaflets. The two types of nanovesicles have somewhat different transbilayer asymmetries, see the two leftmost data sets in Fig. 2, with a highly stretched outer leaflet, arising from a significant positive outer leaflet tension $\Sigma_{ol}$, and with a highly compressed inner leaflet, corresponding to a significant negative inner leaflet tension $\Sigma_{il}$. For a bilayer tension close to zero, both types of nanovesicles have the same volume $v = v_0 = 0.97$ and the same diameter of 37 nm. When a small $\alpha$ droplet with a diameter of 11.2 nm adheres to such a nanovesicle, a reduction of the vesicle volume to $v$ = 0.6 leads to the shape transformations displayed in Fig. 3. Thus, the endocytic process shown in this figure is driven by the combined action of droplet adhesion and volume reduction. The details of our protocol for volume reduction, which mimics the experimental procedure of osmotic deflation, are described in the "Methods" section after Eq. (2).

In Fig. 3a, the vesicle membrane contains $N_{ol} = 5400$ lipids in its outer and $N_{il} = 4700$ lipids in its inner leaflet. The figure panel displays both the top view and the half cut view of the vesicle–droplet couple for four different time points, starting at $t = 0\,\mu s$ and ending at $t = 10\,\mu s$ for volume $v = 0.6$. A more detailed view of this shape evolution is provided by the time-lapse Movies S1 and S2 in the Supplementary Information (SI). Inspection of the top and half cut views in Fig. 3a shows that the droplet is partially engulfed at $t = 0\,\mu s$ and completely engulfed at $t = 0.3\,\mu s$. For partial engulfment, the membrane forms an open neck, whereas complete engulfment implies a closed membrane neck. Furthermore, the time evolution of the vesicle–droplet couple proceeds via axisymmetric shapes with a circular contact line, which leads to fission of the membrane neck at $t = 2\,\mu s$, thereby generating a small intraluminal vesicle that encloses the droplet.

In Fig. 3b, the vesicle membrane contains $N_{ol} = 5500$ lipids in its outer and $N_{il} = 4600$ lipids in its inner leaflet, which differs from the nanovesicle in Fig. 3a by reshuffling 100 lipids from the inner to the outer leaflet. In Fig. 3b, we display again both the top view and the half cut view of the vesicle–droplet morphology for four different time points, starting at $t = 0\,\mu s$ and ending at $t = 12\,\mu s$, while keeping the volume fixed at $v = 0.6$. A more detailed view of this shape evolution is provided by the time-lapse Movies S3 and S4. We again observe partial engulfment of the droplet at $t = 0\,\mu s$ whereas complete engulfment is reached somewhat later at $t = 4\,\mu s$. Furthermore, the time evolution of the vesicle–droplet couple again proceeds via axisymmetric shapes with a circular contact line but the membrane neck adjacent to this contact line now undergoes fission at the later time $t = 12\,\mu s$.

The two nanovesicles in Fig. 3a, b have a different transbilayer asymmetry as encoded in the different lipid numbers $N_{ol}$ and $N_{il}$. Comparison of the snapshots in the two panels shows that a smaller asymmetry as in Fig. 3b delays the onset of complete engulfment as well as the fission of the membrane neck. However, in both cases, the shape evolution proceeds in an axisymmetric manner and leads to the fission of the membrane neck, thereby generating two nested daughter vesicles. This endocytic process resembles fluid endocytosis or pinocytosis, a process of cellular uptake by which fluid droplets in the extracellular medium are engulfed by the plasma membrane and the resulting membrane neck undergoes fission, thereby forming an intraluminal vesicle for intracellular transport[54–60].

### Nonaxisymmetric engulfment without membrane fission

Next, we study nanovesicles with $N_{ol} = 5700$ and $N_{ol} = 5963$ lipids in their outer leaflets, corresponding to the two rightmost data sets in Fig. 2. These vesicles are formed by bilayer membranes with a reduced transbilayer stress asymmetry compared to those with $N_{ol} = 5400$ and $N_{ol} = 5500$ in Fig. 3. This further reduction of the stress asymmetry now leads to a qualitatively different shape evolution of the vesicle–droplet couples when we deflate the vesicles, using again the volume reduction protocol as described in the "Methods" section. Indeed, as shown in Fig. 4, this shape evolution now involves a transition to nonaxisymmetric shapes with noncircular contact lines and noncircular membrane necks without fission. In the top rows of Fig. 4a, b, the contact lines are enframed by white dashed circles and rectangles to emphasize the different contact line shapes. Furthermore, it is instructive to look at the nonaxisymmetric shapes in Fig. 4 from different angles, see Figs. S1 and S2.

The strongly noncircular shape of the contact line implies that the vesicle membrane adjacent to this line forms a closed neck with a tight-lipped shape[51]. These unusual shapes of contact line and membrane neck remain stable until $30\,\mu s$, preventing the fission of the necks and, thus, the formation of intraluminal vesicles. As will become clear further below, the strongly noncircular shape of the contact line and the tight-lipped shape of the membrane neck are caused by negative values of the contact line tension.

### Engulfment of droplets with different sizes

In order to determine the effect of droplet size on the engulfment process, we also studied the vesicle–droplet morphologies for larger droplets with diameter $18.7d$ or 15 nm and $24.5d$ or 19.6 nm. In Fig. 5, we display the shape evolution of vesicle–droplet couples for the same vesicle size of 37 nm as before but for a larger droplet with a diameter of 19.6 nm. The large droplet size prevents the complete engulfment of the droplet but we observe different patterns of partial engulfment depending on the transbilayer asymmetry as encoded by the lipid number $N_{ol}$. For $N_{ol} = 5400$ as shown in Fig. 5a, the contact line remains circular and the vesicle–droplet couple remains axisymmetric as we reduce the vesicle volume to $v = 0.45$. In contrast, for $N_{ol} = 5700$ as shown in Fig. 5b, the contact line becomes noncircular and the vesicle–droplet couple becomes nonaxisymmetric for the same volume reduction. Taken together, our simulations show that the process of droplet engulfment by the membrane of a nanovesicle depends on the transbilayer asymmetry of the membrane and on the relative size of droplet and vesicle.

### Three different surface tensions

A general feature of the vesicle–droplet morphologies is that the contact line between the droplet and the nanovesicle partitions the vesicle membrane into two segments, the membrane segment $\alpha\gamma$ adjacent to the $\alpha$ droplet and the membrane segment $\beta\gamma$ in contact with the aqueous bulk phase $\beta$, see Fig. 1c. As a consequence, the force balance along the contact line involves three different surface tensions, the interfacial tension $\Sigma_{\alpha\beta}$, which exerts capillary forces onto the contact line, as well as the two bilayer tensions $\Sigma_{\alpha\gamma}$ and $\Sigma_{\beta\gamma}$ acting within the $\alpha\gamma$ and the $\beta\gamma$ membrane segment, respectively. Furthermore, this force balance also depends on the line tension of the contact line itself[18,23].

In order to understand the consequences of the force balance at the contact line, we first consider the interfacial tension $\Sigma_{\alpha\beta}$ and the bilayer tensions $\Sigma_{\alpha\gamma}$ and $\Sigma_{\beta\gamma}$ of the two membrane segments for different stress asymmetries as controlled by the lipid number $N_{ol}$. Using the computational approach described in the "Methods" section, we determine these three surface tensions as a function of $N_{ol}$ for different droplet sizes. The results of this computation are displayed in Fig. 6 for nanovesicles with adjusted volumes $v = v_0$. This figure consists of three panels corresponding to three different droplet diameters as given by $14d$ or 11.2 nm, $18.7d$ or 15 nm, and $24.5d$ or 19.6 nm.

Inspection of Fig. 6 shows that the value of the interfacial tension $\Sigma_{\alpha\beta}$ is essentially independent of both the droplet size and the transbilayer stress asymmetry. In contrast, the bilayer tensions of the two membrane segments vary significantly with the lipid number $N_{ol}$ and, thus, with the stress asymmetry. Indeed, for all three droplet sizes in Fig. 6a, b, and c, the segment tension $\Sigma_{\alpha\gamma}$ (red data) increases linearly with lipid number $N_{ol}$ whereas the segment tension $\Sigma_{\beta\gamma}$ (blue data) decreases linearly with $N_{ol}$. The corresponding linear fits are depicted as red and blue broken lines in Fig. 6. Thus, as we increase $N_{ol}$ by moving a certain number of lipids from the inner to the outer leaflet, we increase the segment tension $\Sigma_{\alpha\gamma}$ of the membrane segment $\alpha\gamma$ adjacent to the $\alpha$ droplet and simultaneously decrease the segment tension $\Sigma_{\beta\gamma}$ of the $\beta\gamma$ segment in contact with the aqueous bulk phase $\beta$.

The observed trend for the dependence of the segment tensions on the lipid number $N_{ol}$ can be understood as follows. As we increase the lipid number $N_{ol}$, by reshuffling lipids from the inner to the outer leaflet for constant total lipid number $N_{ol} + N_{il}$, the membrane prefers to attain a more positive curvature. This curvature preference is in line with the positive curvature of the $\beta\gamma$ membrane segment but antagonistic to the negative curvature of the $\alpha\gamma$ membrane segment, see Fig. 1c. Therefore, with increasing $N_{ol}$, the $\beta\gamma$ segment becomes more relaxed, which implies a reduced membrane tension $\Sigma_{\beta\gamma}$, whereas the $\alpha\gamma$ membrane becomes more stretched and thus is subject to an increased tension $\Sigma_{\alpha\gamma}$.

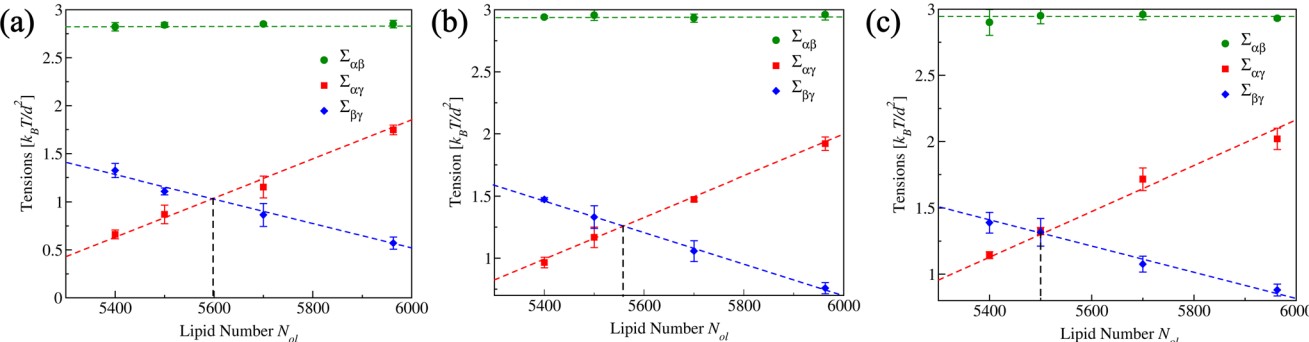

**Fig. 6 | Interfacial tension $\Sigma_{\alpha\beta}$ (green circles), bilayer tension $\Sigma_{\alpha\gamma}$ (red squares) of the membrane segment $\alpha\gamma$, and bilayer tension $\Sigma_{\beta\gamma}$ (blue diamonds) of the membrane segment $\beta\gamma$ as functions of the lipid number $N_{ol}$ in the outer bilayer leaflet.** All nanovesicles have the same volume $v = v_0$. The data in **a–c** are obtained for the three droplet diameters 11.2 nm, 15 nm, and 19.6 nm, respectively. Each data point represents the mean value over $n = 15$ statistically independent samples obtained from three replicated simulations. Each error bar is the SEM. The largest surface tension is provided by the interfacial tension $\Sigma_{\alpha\beta}$, which is essentially

independent of both the lipid number $N_{ol}$ and the droplet size. For all three droplet sizes, the segment tension $\Sigma_{\alpha\gamma}$ increases linearly and the segment tension $\Sigma_{\beta\gamma}$ decreases linearly as we increase the lipid number $N_{ol}$. The numerical values of the different surface tensions are provided in Table 1. For each droplet size, linear interpolation (broken lines) then leads to a characteristic value of $N_{ol}$, denoted by $N_{ol}^{=}$, with identical membrane tensions $\Sigma_{\alpha\gamma} = \Sigma_{\beta\gamma}$ of the two membrane segments. The numerical values of $N_{ol}^{=}$ depend on the droplet size, see penultimate column of Table 1.

One remarkable feature of this behavior is that all three red lines in Fig. 6a–c have essentially the same positive slope $d\Sigma_{\alpha\gamma}/dN_{ol}$ and that all three blue lines in this figure have essentially the same negative slope $d\Sigma_{\beta\gamma}/dN_{ol}$. Furthermore, for each droplet size, the red and blue fitting lines intersect each other at a certain characteristic lipid number $N_{ol}^{=}$, for which both segments experience the same membrane tension, i.e., for which $\Sigma_{\alpha\gamma} = \Sigma_{\beta\gamma}$. The numerical values of $N_{ol}^{=}$, which depend on the droplet size, are given in the penultimate column of Table 1.

It is interesting to note that for $N_{ol} = 5400$ and 5500, which are both smaller than $N_{ol}^{=}$, the vesicle–droplet morphology remains axisymmetric during the whole engulfment process, which is directly apparent from the circular contact lines in Fig. 3. Furthermore, for $N_{ol} = 5400$ and 5500, complete engulfment leads to a circular membrane neck that undergoes fission, thereby generating a small intraluminal daughter vesicle. On the other hand, for $N_{ol} = 5700$ and 5963, which are both larger than $N_{ol}^{=}$, the rotational symmetry of the vesicle–droplet morphology breaks down during the engulfment process and the contact line between the $\alpha$ droplet and the vesicle membrane attains a tight-lipped geometry, see Fig. 4. The tendency to break the axial symmetry increases for larger values of $N_{ol}$ as can be concluded from the shape evolution in Fig. 4b for $N_{ol} = 5963$ because, in the latter case, both contact line and membrane neck acquire an elongated, noncircular shape even before the droplet is completely engulfed.

**Line tension of contact line between droplet and membrane**
To get additional insight into the transition from the axisymmetric morphologies for $N_{ol} < N_{ol}^{=}$ to the nonaxisymmetric morphologies for $N_{ol} > N_{ol}^{=}$, we now consider the line tension of the contact line. As mentioned before, the vesicle–droplet couple with volume $v = v_0$ has an axisymmetric shape and a circular contact line for all studied values of $N_{ol}$. For such an axially symmetric shape, we can determine the line tension $\lambda$ from the force balance along the contact line as described by Eq. (12) in the "Methods" section. As a result, we obtain the line tension displayed in Fig. 7 for different lipid numbers $N_{ol}$. The numerical values of these line tensions are included in Table 1.

One remarkable property of the line tension is that it changes sign as we vary the lipid number $N_{ol}$. As shown in Fig. 7, the line tension is positive for $N_{ol} = 5400$ and 5500 but negative for $N_{ol} = 5700$ and 5963. A positive line tension favors a circular contact line adjacent to an axisymmetric membrane neck as in Fig. 3. When the droplet becomes completely engulfed by the membrane, the membrane neck remains circular, closes in an axisymmetric manner, and subsequently

undergoes cleavage, thereby generating a smaller daughter vesicle that contains the droplet. In contrast, a negative line tension favors a noncircular contact line together with a nonaxisymmetric membrane neck as in Fig. 4. When the latter neck is closed, it assumes an elongated, tight-lipped shape. This unusual behavior can be understood as follows. The contribution of the contact line to the free energy of the vesicle–droplet system is equal to $\lambda L_{cl}$, with the line tension $\lambda$ and the length $L_{cl}$ of the contact line. A negative line tension implies that this free energy contribution is negative as well and that the contact line would like to increase its length $L_{cl}$. At the same time, the system would also like to reduce the area of the $\alpha\beta$ interface, which is bounded by the contact line. Thus, the system tries to maximize the length of the contact line and to simultaneously minimize the area of the $\alpha\beta$ interface. Both requirements can be satisfied by a noncircular, strongly elongated shape of the contact line. However, as the contact line becomes further elongated, the adjacent membrane segment become more strongly curved with a concomitant increase of its bending energy. The interplay between the free energy of the contact line and the bending energy of the adjacent membrane segment determines the overall extension of the contact line and the tight-lipped neck.

As shown in Fig. 7, linear interpolation of the data now leads to a characteristic value $N_{ol} = N_{ol}^{[0]}$, at which the line tension vanishes. The numerical values of these lipid numbers are provided in the last column of Table 1. Comparison of the lipid numbers $N_{ol}^{[0]}$ with the lipid numbers $N_{ol}^{=}$, for which the two membrane segments experience the same bilayer tension, see penultimate column of Table 1, shows that the two numbers are equal to each other, within the accuracy of our computations, for all droplet sizes.

## Discussion
In this paper, the endocytosis of liquid droplets and molecular condensates has been addressed by studying their interactions with nanovesicles. As a result, we demonstrated that the vesicle–droplet systems follow different endocytic pathways at the nanoscale. Sufficiently small droplets are completely engulfed by the membrane which then forms a closed membrane neck, following two different pathways. One pathway is provided by axisymmetric shapes of vesicle and droplet as well as subsequent fission of the circular membrane neck which generates two nested daughter vesicles (Fig. 3 as well as Movies S1–S4). Another pathway leads to nonaxisymmetric shapes and noncircular membrane necks which impede the fission process (Fig. 4 as well as Movies S5–S8). Larger droplets are only partially engulfed by the membrane which now forms an open membrane neck, following

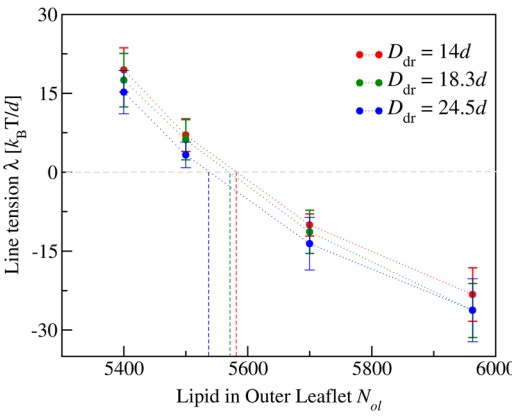

**Fig. 7 | Line tension $\lambda$ of contact line between droplet and vesicle membrane as a function of lipid number $N_{ol}$ for three different droplet sizes $D_{dr}$, calculated via the force balance relationship in Eq. (12).** As we increase $N_{ol}$, which encodes the transbilayer asymmetry, the line tension undergoes a transition from positive to negative values. The line tension is positive for $N_{ol} = 5400$ and 5500, for which the whole engulfment process remains axisymmetric and leads to neck cleavage and membrane fission. On the other hand, for $N_{ol} = 5700$ and 5963, the line tension has a negative value and leads to the formation of a tight-lipped membrane neck as in Fig. 4 which prevents the fission process. The dashed vertical lines provide estimates for the lipid numbers $N_{ol} = N_{ol}^{[0]}$ which lead to $\lambda = 0$ for the different droplet sizes $D_{dr}$. The numerical values of $N_{ol}^{[0]}$, which depend on the droplet size, are given in the last column of Table 1. Each red, green, and blue data point represents the mean value over $n = 15$ statistically independent samples obtained from three replicated simulations. Each error bar is the SEM.

again two pathways with circular and noncircular necks (Fig. 5). Which pathway is taken depends on the droplet size and on two key parameters: (i) the transbilayer stress asymmetry (Fig. 2) between the two leaflets of the bilayer membrane in the absence of a droplet, as controlled by the outer lipid number $N_{ol}$ during the initial assembly of the vesicle; and (ii) the line tension of the contact line between droplet and vesicle membrane (Fig. 7).

The division of a nanovesicle into two daughter vesicles is observed for positive line tension of the contact line and for sufficiently large stress asymmetries between the two bilayer leaflets. A positive value of the contact line tension is necessary for the nanovesicle to form a circular membrane neck which represents a prerequisite for neck fission. The transbilayer stress asymmetry plays the same role for nanovesicles as the spontaneous curvature for giant vesicles. In the latter case, the theory of curvature elasticity predicts that a sufficiently large spontaneous curvature generates a strong constriction force at the membrane neck that is sufficient to cleave the neck[61] as has been observed experimentally for giant unilamellar vesicles[62]. Our simulation study demonstrates an analogous fission mechanism for nanovesicles, with neck cleavage and vesicle division being induced by a sufficiently large transbilayer stress asymmetry.

Our study also revealed that the lipid number $N_{ol}$ represents a key parameter for the bilayer tensions of the two membrane segments $\alpha\gamma$ and $\beta\gamma$, which arise from the partitioning of the vesicle membrane by the contact line between membrane and droplet (Fig. 1). This partitioning leads to the membrane segment $\alpha\gamma$, which is in contact with the droplet, and to the complementary segment $\beta\gamma$, which is not exposed to the droplet. As shown in Fig. 6, the segment tension $\Sigma_{\alpha\gamma}$ increases linearly whereas the segment tension $\Sigma_{\beta\gamma}$ decreases linearly with increasing $N_{ol}$. For all droplet sizes, the two segment tensions become equal for a certain lipid number $N_{ol} = N_{ol}^{=}$, for which the whole membrane of the nanovesicle is subject to the same bilayer tension $\Sigma_{\alpha\gamma} = \Sigma_{\beta\gamma}$. Within the accuracy of our computational approach, this boundary value is equal to lipid number $N_{ol} = N_{ol}^{[0]}$, for which the line tension of the contact line vanishes (Fig. 7 and Table 1).

For computational convenience, we considered here relatively simple vesicle–droplet systems based on bilayer membranes with a single lipid component and on droplets arising from liquid–liquid phase separation in a binary liquid mixture. However, our computational approach can be directly generalized to more complex vesicle–droplet systems consisting of bilayer membranes with several molecular components and of multicomponent liquid mixtures that form biomolecular condensates. The description of these more complex systems involves additional molecular interactions as well as additional parameters that describe the composition of bilayers and liquid mixtures. In spite of this increased complexity, all tensions considered here for the simple systems remain well-defined and useful nanoscopic quantities for the more complex systems. Furthermore, these tensions are expected to fulfill very similar relationships as found here for the simple systems.

Indeed, for tensionless bilayers, the two leaflet tensions must again lead to one stretched and one compressed leaflet as in Fig. 2, irrespective of the bilayer's molecular composition. Likewise, when the nanovesicle interacts with a small droplet, the bilayer tensions of the two membrane segments will again follow the general trend displayed in Fig. 6: As we reshuffle lipids from the inner to the outer leaflet, the vesicle membrane prefers to attain a more positive mean curvature which implies (i) an increased bilayer tension $\Sigma_{\alpha\gamma}$ for the $\alpha\gamma$ segment in contact with the droplet and (ii) a reduced bilayer tension $\Sigma_{\beta\gamma}$ for the $\beta\gamma$ segment not exposed to the droplet. As a consequence, any vesicle–droplet system should exhibit a special state with identical segment tensions $\Sigma_{\alpha\gamma} = \Sigma_{\beta\gamma}$, corresponding to the intersection points in Fig. 6.

In the present simulation study, we used coarse-grained molecular dynamics in the form of Dissipative Particle Dynamics[31,32] to reach the length and time scales that are relevant for the remodeling of nanovesicles. It will be relatively straightforward to perform analogous simulations based on other coarse-grained modeling approaches such as the widely used Martini force field[63]. Likewise, atomistic modeling could be used to obtain a more specific description of the molecules and molecular interactions. However, studying droplet endocytosis using such atomistic models will be computationally rather expensive, at least for the time being. In addition, it has been found difficult to compute stress profiles and membrane tensions for molecular models with atomistic resolution. On the other hand, our protocol to assemble the bilayer leaflets does not require that we actually measure these tensions. Indeed, using again the lipid numbers $N_{ol}$ and $N_{il}$ assembled in the outer and inner bilayer leaflets as control parameters, we will be able to observe the different pathways of droplet endocytosis in atomistic simulations as well.

For the lipid bilayers studied here, the lipid molecules between the two bilayer leaflets did not undergo flip-flops on the time scales of our simulations. This suppression of flip-flops agrees with experimental observations on phospholipids with measured flip-flop times that typically exceed several minutes and often are of the order of many hours[64,65]. In the simulations, flip-flops become more frequent when cholesterol is added to the lipid bilayers[66] or when the bilayer leaflets are exposed to relatively large stress asymmetries[67].

Finally, the endocytic pathways described here should also be accessible to experimental studies on synthetic nanovesicles or liposomes. For such vesicles. a variety of experimental protocols has been developed to change the molecular composition of their bilayer leaflets. These protocols include: a post-insertion method by fusing the vesicles with micelles that carry ligand molecules[68]; cyclodextrin-mediated exchange of lipids between two vesicle populations[69]; flip-flops of an anionic phospholipid from the outer to the inner leaflet driven by $Ca^{2+}$ ions in the exterior solution[70,71]; and enzymatic conversion of phospholipid headgroups in the outer leaflet[72,73]. These processes change the molecular composition in the bilayer leaflets and will typically change the leaflet tensions as well. When such changes are

**Table 2 | DPD force parameters $f_{ij}$ in units of $k_B T/d$**

| $f_{ij}$ | $j = H$ | $j = C$ | $j = W$ | $j = S$ |
|---|---|---|---|---|
| $i = H$ | 30 | 50 | 25 | 25 |
| $i = C$ | 50 | 10 | 75 | 75 |
| $i = W$ | 25 | 75 | 25 | 70 |
| $i = S$ | 25 | 75 | 70 | 25 |

The system is built up from lipid head (H), lipid chain (C), water (W), and solute (S) beads.

combined with osmotic deflation to reduce the vesicle volume, synthetic liposomes in contact with small liquid droplets are likely to undergo different pathways of endocytosis as observed here in silico.

## Methods

### Coarse-grained molecular dynamics

We use coarse-grained molecular dynamics simulations to study nanovesicles interacting with small liquid droplets. The nanovesicle have a fixed diameter of $46.6d$ or 37 nm but differ in their transbilayer asymmetry, see Fig. 2. The small droplets, which are formed by liquid–liquid phase separation in the exterior aqueous solution, have a diameter between 11 nm and 19 nm. The equilibration of these systems requires simulation times of many microseconds. To gain access to these length and time scales, we use a coarse-grained molecular model which we study by dissipative particle dynamics[31,32]. Our molecular system is built up from four types of beads that represent small molecular groups: water (W) beads, lipid chain (C) beads, lipid head (H) beads, as well as solute (S) beads. The lipid molecules have a head group consisting of three H beads and two hydrocarbon chains, each of which consists of six C beads. All beads have the same diameter $d$, corresponding to about 0.8 nm. Each pair of beads interacts with short-ranged pairwise additive forces as described previously[49], including the conservative force

$$\vec{F}_{ij}^C = f_{ij}\left(1 - r_{ij}/d\right)\hat{r}_{ij}, \quad \text{for } r_{ij} < d$$
$$= 0 \quad \text{for } r_{ij} > d \tag{1}$$

with the force parameter $f_{ij}$, the unit vector $\hat{r}_{ij}$ pointing from bead $j$ to bead $i$, and the distance $r_{ij}$ between bead $i$ and bead $j$. The numerical values of the force parameters $f_{ij}$ are displayed in Table 2. The lipid head (H) and chain (C) beads experience the same interactions with the water beads and the solute beads, i.e., $f_{HW} = f_{HS}$ and $f_{CW} = f_{CS}$. All force parameters $f_{ij}$ have the same values as in Ref. [51], apart from $f_{WS}$, which is chosen to be $f_{WS} = 70$. The latter, relatively high value of $f_{WS}$ ensures that the binary mixture of W and S beads undergoes phase separation for relatively small solute concentrations, see Eq. (6) further below. The choice $f_{HW} = f_{HS}$ and $f_{CW} = f_{CS}$ is made to reduce the number of parameters that define our simulation model. This choice implies that the water and solute beads have the same affinity to the lipid bilayer and that the contact angle is close to 90°, see Table 2. In order to describe a preferential binding of the solute to the membrane, we would have to consider an affinity contrast between the S and W beads as has been previously explored for DPD simulations of planar lipid bilayers[51]. Such an affinity contrast would also provide a simple model system for the binding of individual proteins to lipid bilayers[27,35].

Our simulations were performed using LAMMPS (Large Scale Atomic/Molecular Massively Parallel Simulator) which is an efficient and parallelized classical molecular dynamics simulator. We study a cuboid simulation box with all three sides being equal to $80d$ which implies the box volume $(80d)^3$. The total number of beads inside the box is 1,591,900. The bulk water density is kept fixed at $\rho = 3/d^3$, which ensures the bulk pressure to have the standard DPD value

$P = 20.7 k_B T/d^3$ arising solely from the bead-bead interactions, i.e., without the constant contribution of $3\,k_B T/d^3$ from the kinetic energy[31].

### Initial assembly of nanovesicles

The simulation setup starts with the construction of spherical vesicles by assembling $N_{il}$ lipids in the inner leaflet and $N_{ol}$ lipids in the outer leaflet of each vesicle. We vary the two lipid numbers $N_{il}$ and $N_{ol}$ but keep the total number of lipids, $N_{il} + N_{ol}$, equal to 10,100. Therefore, changes in $N_{ol}$ are obtained by reshuffling lipids between the two leaflets: The outer lipid number $N_{ol}$ is increased by moving lipids from the inner to the outer leaflet and decreased by moving lipids from the outer to the inner leaflet. In the present study, we focussed on nanovesicles with four different transbilayer asymmetries as generated by four different values of $N_{ol}$ as given by $N_{ol} = 5400$, 5500, 5700, and 5963. The largest value $N_{ol} = 5963$ corresponds to nanovesicles with two tensionless leaflets, see rightmost data point in Fig. 2. For all $N_{ol}$-values studied here, the initial assembly leads to nanovesicles with a diameter of $46.6d$ or 37 nm, corresponding to the diameter of the outer head group layer.

### Volume of nanovesicles

The inner head group layer of the vesicle membrane encloses the interior aqueous solution, which contains $N_W^{in}$ water beads and no solute beads. After the initial assembly of the spherical vesicle, it encloses $N_W^{in} = N_W^{isp} = 90,400$ water beads. To measure the vesicle volume, we use the dimensionless volume parameter

$$\nu \equiv \frac{N_W^{in}}{N_W^{isp}} \quad \text{which satisfies } 0 < \nu \le 1. \tag{2}$$

Thus, after the initial assembly of the nanovesicle, the volume parameter has the value $\nu = 1$, and any volume reduction leads to $N_W^{in} < N_W^{isp}$ and $\nu < 1$. To reduce the vesicle volume, we extend the protocol developed in refs. [33,49]. More precisely, the reduction of the vesicle volume, which mimics osmotic deflation, is performed as follows. We start from the initially assembled nanovesicle with volume $\nu = 1$ and reduce this volume in discrete steps of $\Delta \nu = 0.1$, by moving subvolumes of water beads from the interior to the exterior solution. After each volume reduction step, the vesicle–droplet couple is equilibrated for 10–20 µs. The resulting sequence of volume reduction steps is terminated when we reach a volume $\nu$ for which the equilibrated droplet is completely engulfed by the vesicle membrane.

To avoid bilayer rupture, we are primarily interested in nanovesicles with tensionless bilayers. In order to reach such a tensionless bilayer state, we need to measure the bilayer tension $\Sigma$ as described in the next paragraph. We then find that a relatively small volume reduction is sufficient to reach the specific volume $\nu = \nu_0$, for which the bilayer membrane becomes tensionless and the bilayer tension $\Sigma$ becomes close to zero. Indeed, for all nanovesicles studied here, the bilayer becomes tensionless when the vesicle volume is reduced from its initial value $\nu = 1$ by less than one percent, see Table S1.

### Bilayer tension and leaflet tensions of spherical nanovesicles

The lipid numbers $N_{ol}$ and $N_{il}$ assembled in the outer and inner leaflets determine the leaflet tensions $\Sigma_{ol}$ and $\Sigma_{il}$ experienced by these leaflets as well as the bilayer tension $\Sigma = \Sigma_{ol} + \Sigma_{il}$. To compute these tensions, we apply the computational protocol developed in ref. [49]. We consider the normal and tangential components of the pressure tensors, $P_N(r)$ and $P_T(r)$ as well as the stress profile $s(r)$ across the bilayer, which is defined by

$$s(r) \equiv P_N(r) - P_T(r) \tag{3}$$

using the computational method in ref. [74]. The bilayer tension $\Sigma$ can then be obtained by integrating this stress profile over the radial

coordinate $r$ which leads to

$$\Sigma = \int_0^\infty dr\,[P_N(r) - P_T(r)] = \int_0^\infty dr\,s(r), \qquad (4)$$

in close analogy to the interfacial tension[75] of a spherical liquid droplet. Furthermore, the two leaflet tensions are obtained via

$$\Sigma_{il} = \int_0^{R_{mid}} dr\,s(r) \quad \text{and} \quad \Sigma_{ol} = \int_{R_{mid}}^\infty dr\,s(r) \qquad (5)$$

where $R_{mid}$ is the radius of the bilayer's midsurface. In principle, there are several possible procedures to define the radius $R_{mid}$ but, in practice, all of these procedures give very similar values for $R_{mid}$[49]. Here, we calculate this radius by locating the midsurface at the peak of the density profile $\rho_C$ of the lipid chain (C) beads.

For each value of $N_{ol}$, we considered several nanovesicles with different volumes $v \lesssim 1$ and calculated the bilayer tension $\Sigma$ of the vesicle membranes via Eq. (4). In this way, we identified the specific volume $v = v_0$, for which the bilayer tension is close to zero. After the volume $v_0$ has been determined, we used Eq. (5) to compute the two leaflet tensions. The results of these calculations are displayed in Fig. 2 and Table S1.

In each case, we create a single small droplet of $\alpha$ phase inside the $\beta$ phase, initially not in contact with the nanovesicle. The protocol starts with the assembly of all solute beads into a spherical subvolume, which forms an initially water-free spherical droplet. Equilibration of this droplet then leads to the uptake of a few water beads and to the formation of an equilibrated $\alpha$ droplet coexisting with the bulk phase $\beta$. As long as the droplet is not in contact with the nanovesicle, it has a spherical shape as in Fig. 1a. The diameter of the spherical droplet increases with the solute concentration and is equal to $14\,d$ or 11.2 nm for $\Phi_S = 0.004$, $18.7\,d$ or 15 nm for $\Phi_S = 0.006$, and $24.5\,d$ or 19.6 nm for $\Phi_S = 0.009$. The corresponding number $N_S$ of solute beads is given by $N_S = 5400$ for $\Phi_S = 0.004$, $N_S = 8100$ for $\Phi_S = 0.006$ and $N_S = 12180$ for $\Phi_S = 0.009$, respectively.

Next, in order to observe endocytic engulfment, we bring each droplet in contact with the nanovesicle, and perform further equilibration of the systems by reducing the vesicle volume $v$.

## Liquid−liquid phase separation in the exterior aqueous solution

To create small liquid droplets, we consider an exterior aqueous solution consisting of $N_W^{ex}$ water beads and $N_S$ solute beads. The phase diagram of this binary mixture contains a two-phase coexistence region, in which the mixture undergoes liquid−liquid phase separation into a solute-rich liquid phase $\alpha$ and a solute-poor liquid phase $\beta$. For small solute concentrations, the $\alpha$ phase forms a small droplet as in Fig. 1. The mixture's phase diagram is defined by the solute concentration and by the solute's solubility in water[33]. The solubility $\zeta$ is given by the ratio $(f_{WW} + f_{SS})/(2f_{WS})$ of the DPD force parameters and has here the constant value $\zeta = 50/140 = 0.357$ as follows from Table 2. The solute concentration is measured by the solute mole fraction

$$\Phi_S \equiv \frac{N_S}{N_S + N_W^{ex}} \quad \text{with } 0 \le \Phi_S \le 1. \qquad (6)$$

We studied three solute concentrations corresponding to $\Phi_S = 0.004$, 0.006, and 0.009. In each case, we create a single small droplet of $\alpha$ phase inside the $\beta$ phase, initially not in contact with the nanovesicle[33].

The protocol starts with the assembly of all solute beads into a spherical subvolume, which forms an initially water-free spherical droplet. Equilibration of this droplet then leads to the uptake of a few water beads and to the formation of an equilibrated $\alpha$ droplet coexisting with the bulk phase $\beta$. As long as the droplet is not in contact with the nanovesicle, it has a spherical

shape as in Fig. 1a. The diameter of the spherical droplet increases with the solute concentration and is equal to $14\,d$ or 11.2 nm for $\Phi_S = 0.004$, $18.7\,d$ or 15 nm for $\Phi_S = 0.006$, and $24.5\,d$ or 19.6 nm for $\Phi_S = 0.009$. The corresponding number $N_S$ of solute beads is given by $N_S = 5400$ for $\Phi_S = 0.004$, $N_S = 8100$ for $\Phi_S = 0.006$ and $N_S = 12180$ for $\Phi_S = 0.009$, respectively. In order to observe endocytic engulfment, we bring the droplets in contact with the nanovesicles and perform further equilibration of the systems after reducing the vesicle volume $v$.

## Maximal droplet size for complete engulfment

For any closed surface in three-dimensional space, the surface area $A$ and the enclosed volume $V$ satisfy the isoperimetric inequality[52,53]

$$A^3 \ge 36\pi V^2. \qquad (7)$$

The limiting case $A^3 = 36\pi V^2$ applies to a spherical shape, which provides the shape with the smallest possible surface area $A$ for a given volume $V$. We now apply this inequality to the membrane shape of a completely engulfed droplet as displayed in Figs. 3 and 4. More precisely, we apply the inequality to the shape of the inner leaflet of these membrane shapes, which enclose the $\alpha$ droplet and the interior volume of the vesicle which is filled with the liquid phase $\gamma$, compare Fig. 1. For a completely engulfed droplet, the shape of the inner leaflet consists of two segments corresponding to the $\alpha\gamma$ and $\beta\gamma$ membrane segments, which are connected by a closed membrane neck. The inner leaflet of the $\alpha\gamma$ segment has the surface area $A_{\alpha\gamma}$ and encloses the droplet volume $V_\alpha$ which implies the isoperimetric inequality

$$A_{\alpha\gamma} \ge (36\pi)^{1/3} V_\alpha^{2/3}. \qquad (8)$$

Likewise, the inner leaflet of the $\beta\gamma$ segment has surface area $A_{\beta\gamma}$ and encloses the combined volume $V_\alpha + V_\gamma$, where $V_\gamma$ is the volume of the $\gamma$ phase. The corresponding isoperimetric inequality now has the form

$$A_{\beta\gamma} \ge (36\pi)^{1/3} \left(V_\alpha + V_\gamma\right)^{2/3}. \qquad (9)$$

For the total surface area $A = A_{\alpha\gamma} + A_{\beta\gamma}$ of the inner leaflet, we then obtain

$$A = A_{\alpha\gamma} + A_{\beta\gamma} \ge (36\pi)^{1/3}\left[V_\alpha^{2/3} + \left(V_\alpha + V_\gamma\right)^{2/3}\right] \ge 2(36\pi)^{1/3} V_\alpha^{2/3} \qquad (10)$$

where the last inequality follows from $V_\gamma \ge 0$. Therefore, we conclude that complete engulfment of the $\alpha$ droplet is only possible for a sufficiently small droplet volume $V_\alpha$ that satisfies

$$V_\alpha \le \frac{\left(A_{\alpha\gamma} + A_{\beta\gamma}\right)^{3/2}}{2^{3/2}(36\pi)^{1/2}} \equiv V_\alpha^{max} \qquad (11)$$

but impossible for larger droplets with volume $V_\alpha > V_\alpha^{max}$. The boundary case with $V_\alpha = V_\alpha^{max}$ corresponds to two nested spheres formed by the inner leaflet of the bilayer membrane. These two spheres touch each other because the liquid phase $\gamma$ between the $\alpha\gamma$ and the $\beta\gamma$ membrane segments has volume $V_\gamma = 0$.

## Axisymmetric geometry I: computation of three surface tensions

To obtain the bilayer tensions $\Sigma_{\alpha\gamma}$ and $\Sigma_{\beta\gamma}$ of the two membrane segments as well as the interfacial tension $\Sigma_{\alpha\beta}$ of the liquid−liquid interface, we consider a partially engulfed droplet, forming an axisymmetric geometry with the nanovesicle as in Fig. 5a, and apply the

methodology previously developed in ref. [51]. As a result, we obtain the components of the local stress or pressure tensor which now depend on two coordinates, the Cartesian coordinate $y$ along the axis of rotational symmetry and the radial coordinate $r$ perpendicular to this axis. Some examples for the stress components $P_{yy}$ and $\frac{1}{2}(P_{xx}+P_{yy})$ as well as the stress profile $s(r,y) \equiv P_{yy} - \frac{1}{2}(P_{xx}+P_{yy})$ are displayed in Fig. S3 as functions of the two coordinates $y$ and $r$. In the three rows of this figure, we show these local pressures and stresses for vesicles with three different values of $N_{ol}$ as given by (a) $N_{ol} = 5500$ (b) $N_{ol} = 5700$, and (c) $N_{ol} = 5963$. In each case, the volume of the nanovesicle has been adjusted to $v = v_0$. The surface tensions $\Sigma_{\beta\gamma}$, $\Sigma_{\alpha\gamma}$, and $\Sigma_{\alpha\beta}$ are then obtained by integrating each of the stress profiles $s(r,y)$ across the respective surface segment, using an integration path perpendicular to these segments. For convenience, we choose these surface segments to be located close to the axis of rotational symmetry, see the green, red and blue boxes in Fig. S3. The regions within these boxes are discretized using a cubic lattice of mesh points with lattice constant $d$.

## Axisymmetric geometry II: force balance condition

Examples for axisymmetric morphologies of the vesicle–droplet couple are displayed in Fig. 3. The cross-section of such an axisymmetric shape, corresponding to partial engulfment, is schematically shown in Fig. S4. In this case, the contact line is circular and can be characterized by three geometric quantities: the intrinsic contact angle $\theta_\alpha^*$ between the $\alpha\beta$ interface and the $\alpha\gamma$ membrane segment; the tilt angle $\psi_{co}$ of the shape contour at the contact line; and the radius $R_{co}$ of the contact line.

The first variation of the free energy[23] of the vesicle–droplet system with respect to the position of the contact line leads to a force balance of the tangential force components along the contact line. When we take the two $\alpha\gamma$ and $\beta\gamma$ membrane segments to have the same curvature-elastic properties, this force balance leads to the relationship[18,23]

$$\Sigma_{\beta\gamma} - \Sigma_{\alpha\gamma} = \Sigma_{\alpha\beta}\cos(\theta_\alpha^*) + \frac{\lambda}{R_{co}}\cos\psi_{co} \qquad (12)$$

between the bilayer tensions $\Sigma_{\beta\gamma}$ and $\Sigma_{\alpha\gamma}$ of the two membrane segments, the interfacial tension $\Sigma_{\alpha\beta}$, the geometric quantities $\theta_\alpha^*$, $R_{co}$ and $\psi_{co}$, as well as the contact line tension $\lambda$.

The computation of the three surface tensions has been described in the previous paragraph. In addition, we analyze the vesicle–droplet geometry to determine the intrinsic contact angle $\theta_\alpha^*$, the contact line radius $R_{co}$, and the tilt angle $\psi_{co}$ which are defined in Fig. S4. The numerical values of these quantities are provided in Table S2 and plotted in Fig. S5. Finally, we insert the three surface tensions and the three geometric parameters into Eq. (12) to compute the unknown contact line tension $\lambda$.

In order to determine the line tension by the procedure just described, we need to analyze an axisymmetric vesicle–droplet morphology. Such a morphology was found to apply to all values of $N_{ol}$ provided we consider partial engulfment for a sufficiently large vesicle volume. Therefore, even if the vesicle–droplet couple acquires a nonaxisymmetric morphology for sufficiently low vesicle volumes as in Fig. 4, we can restore the axisymmetry by increasing the volume of the vesicle and then apply the analysis based on Eq. (12). In this way, we obtained the negative values of the contact line tension in Fig. 7 and Table 1.

## Reporting summary

Further information on research design is available in the Nature Portfolio Reporting Summary linked to this article.

## Data availability

The raw data underlying the statistics for the mean and SEM values presented in Figs. 2, 6, 7, and S5 are available at https://github.com/rikhiag/DropletEngulfmentData.git. All other data are available from the authors upon request.

## Code availability

Further information on the software used in this study is available in the Nature Research Reporting Summary and in the Nature Research Software Policy linked to this article. The input scripts are available at https://github.com/rikhiag/DropletEngulfment. We used MATLAB software version Matlab_R2021 for post-processing of the stress profiles and density profiles to calculate interfacial tensions as well as leaflet and bilayer tensions.

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

## Acknowledgements

We acknowledge support by the IT team of the Max Planck Institute of Colloids and Interfaces as well as by the Max Planck School Matter to Life, which is funded by the German Federal Ministry of Education and Research (BMBF) in collaboration with the Max Planck Society and the Max Planck Institute of Colloids and Interfaces.

## Author contributions

R.G. and R.L. designed the project. R.G. and V.S. performed the simulations. R.G., V.S., and R.L. analyzed the data and discussed the results. R.L. and R.G. wrote the manuscript.

## Funding

## Competing interests

The authors declare no competing interests.
