## [Peer Review File · Nature Communications]

REVIEWER COMMENTS

Reviewer #1 (Remarks to the Author):

Ghosh and colleagues describe a course-grained computational model for examining the ability of membrane vesicles to engulf liquid-like protein condensates. They vary the size of the condensate and vesicle, as well as the degree of asymmetry in lipid number between the inner and outer leaflet of the vesicle. The results identify limits on the sizes of condensates that can be fully engulfed, showing that symmetric contraction of the membrane neck around the condensate is important for full engulfment and membrane fusion. Interestingly, the model predicts that full fusion of the membrane takes place, resulting in a vesicle-wrapped condensate, encapsulated within the initial vesicle. Overall the work illustrates a potentially interesting mechanism. However, I have a number of questions about its physiological relevance, assumptions, and the chosen parameter variations.

- 1) Very small vesicles (37 nm diameter) were used in the simulations. This represents a very high membrane curvature that is not very characteristic of the plasma membrane. Indeed, most endocytic structures (which have the opposite curvature) are larger than this. I can understand that very small vesicles were used because of the computational cost of simulating larger vesicles. The authors should acknowledge that the present case in non-physiological and comment on how the results might be different if the starting membrane were flatter. Would the same mechanisms be expected to be important?
- 2) How much the droplet spreads on the membrane will be controlled by the surface tension between the droplet and the surrounding solution, and the affinity between the droplet and the membrane surface. The authors seem to have fixed these parameters. How were these values chosen? With the chosen values, is the condensate more liquid-like or solid-like? If liquid-like, how would the viscosity of the condensate compare to literature values from experimental measurements?
- 3) Several groups have recently observed protein droplets to spread to a much greater extent on membrane surfaces, essentially creating a 2D patch on the membrane surface when they encounter it. (Snead and Gladfelter, NCB 2022; Yuan and Stachowiak, PNAS 2021, among several others.) How would that situation be different from what the authors present here? In particular, Yuan et al. showed that these 2D, patch-like condensates also generate inward membrane curvature. How might that mechanism be different (or similar) to the mechanism that the authors are investigating here? The authors should discuss this recent literature and explain how it relates to the present study.
- 4) Along the same lines, a request for clarification: Why do the individual proteins that comprise the droplet not bind to the membrane surface? If the droplet can bind to the membrane, why can't the individual proteins bind? Will this happen if the protein-protein interactions are weakened in comparison to the protein-membrane interactions? Have the authors tried any simulations in that regime? This is an important question in the context of recent findings that membrane-binding may nucleate the assembly of condensates (Snead and Gladfelter, NCB 2022; Day and Stachowiak, NCB 2021).
- 5) I had a hard time understanding the idea of a "tight-lipped shape" to which the authors refer? How exactly is this defined geometrically? Can the authors quantify it in terms of geometrical parameters and report it for the different conditions that they simulated? As it is a major point in the study, I think it should be quantified.
- 6) It was not clear to me why a non-axisymmetric membrane neck corresponds to the "tight-lipped" shape. Can the authors explain?
- 7) How do the authors justify the observed fission/fusion? The driving force for it must come from the adhesion energy between the droplet and the membrane? Is this expected? Why doesn't the membrane simply coat the droplet, rather than undergoing full fission? Full fission implies that the two leaflets are being forced together with sufficient intensity to

remove water molecules and suppress leaflet fluctuations. Would this be expected under physiological conditions or could it be an artifact of the authors' coarse-graining scheme?

Reviewer #2 (Remarks to the Author):

In this manuscript Ghosh and colleagues study the endocytosis process of a liquid droplet by a lipid nanovesicle via molecular dynamics simulations. The authors explore how, depending on the lipid bilayer asymmetry, contact line tension and droplet size, different engulfment pathways arise that may lead to complete endocytosis. In essence, the authors find that there is a critical bilayer asymmetry for which the engulfment process transitions from being completely axisymmetric to adopting a tight-lipped configuration (which prevents inward budding in all cases). As it turns out, this critical point corresponds to vanishing contact line tension and equal membrane tensions in the two segments (in contact with the droplet or in contact with the bulk solution).

This work contributes to the well explored field of theoretical lipid membrane deformations and passive endocytic processes. The main result of the paper is highlighting the role bilayer asymmetry in its functioning. This is interesting, though the conclusion has been put forward before (by the same group see eg Agudo-Canalejo et al, ACS Nano 2015 or Agudo-Canalejo, Soft Matter 2021). I had expected the particle fluidity to play a bigger role in membrane wrapping, similarly to what has been reported before in the context of deformable particles (see eg Shen et al, ACS Nano 2021 for one example), and which would be potentially new. However, the role of the particle fluidity has not been really discussed or flashed out here. Finally, the description of how the simulations have been done is not very clear and leaves the reader with many questions.

Below are detail some more detailed comments and questions:

- I found Fig 3-5 very qualitative and repetitive; they could have been summarized in a single figure. Also, it is unclear how relevant single snapshots are, and how statistically relevant the engulfment times are given that they have been reported for a single system realization. To be convincing, this data should be represented as a plot of quantitative measurements (e.g. times/membrane shapes vs lipid asymmetry) with proper statistical averages.
- The volume control protocols used in the simulations need more clarification. It seems the authors control the vesicle volume during the simulation by varying the number of water molecules enclosed by it. However, the specific way in which this is implemented is not discussed. Also, which kind of a physiological situation this protocol of volume change would correspond to? Shouldn't that happen spontaneously in a simulation instead of being implemented by hand?
- In more details, while the initial reduced volume values $v = v_0$ are explained in detail, the choice of volumes for the engulfment processes (for example $v = 0.6$) and the way this volume change is implemented (is it a sudden change or does it occur over a fixed time interval? If so, what is the deflation rate? How does this compare to deflation rates in experiments?) require an appropriate explanation. Looking at Ref. 39 it seems that in some cases deflation is sufficient to drive endocytosis of the exterior solution. If this is the case, what is the role of the droplet? Is the engulfment driven by the volume reduction or by the droplet or a combination of both? This whole aspect of the simulations needs to be explained in detail.

- The authors explore here the important role of bilayer asymmetry for these engulfment processes by artificially setting a specific asymmetry between the two leaflets which is then kept constant. However, lipid bilayers in nature display reshuffling such as flip-flop events. Even though such processes might not seem to be very rapid in cells, they are known to occur more frequently in coarse-grained molecular dynamics simulations. One wonders how they could affect the results presented here. Can the authors comment on this? Could flip-flops and similar reshuffling events change the asymmetry of the bilayer during the endocytosis and how would that affect the pathways? Have the bilayer been equilibrated before engulfment and the asymmetry measured after equilibration?

- In their introduction the authors make a point of explaining the novelty of using deformable liquid droplets instead of rigid nanoparticles. However, the implications of this deformability for their results remains unclear.

- Specifically, the deformability/fluidity of the droplet is not measured (is it even fluid? This should be measured.) or discussed enough to get an idea of how this setup compares to previous non-deformable nanoparticle engulfment works. Secondly, the deformation of the droplet during the endocytosis process (which seems visible in some snapshots and movies) is never properly characterized and its role is not discussed. Finally, the conclusions of the work seem to be quite independent of the droplet deformability and depend only on the droplet size. How different would these results be with a simple rigid nanoparticle of the same size that has been thoroughly explored before? This requires discussion and comparison to previous work.

Some minor notes:

- The movies would benefit from incorporating more frames to increase the time resolution
- Panel c) in Fig. 1 has a strange bit (like a half-erased bead) towards the top-right corner
- Typo on Page 5: inequality for inequality

Reviewer #3 (Remarks to the Author):

The authors used DPD simulation to explore the mechanism behind vesicles' endocytosis of liquid droplets. The simulation results showed that there were two different endocytic pathways for both partial and complete engulfment. By varying the number of lipids in the outer/inner leaflet and the droplet's volume, the authors obtained two key parameters for endocytic pathways: transbilayer asymmetry of the vesicle membrane and the line tension of the membrane-droplet contact line. The vesicle-droplet systems used in the manuscript are concise and representative, which can reflect the physical relationship in some more complex systems such as the endocytic processes of biomolecular condensates. The reviewer is in support of its publication after the following points are addressed.

1. In terms of interaction force field parameters, whether the interaction parameters of solute beads are adopted for the droplet? If not, what are the interaction parameters for droplets? And why the parameter for solute beads is set to be 70.
2. The diameters of the droplets used in the simulation are 14 d, 18.7 d, and 24.5 d, why did the authors choose the above size? For some droplets with smaller sizes (maybe 8 d, and 10 d), will the complete axisymmetric engulfment happen in the nanovesicles with $N_{ol} = 5700$ or 5963 lipids in the outer lipid bilayer?
3. The author should further analyze why the zip like necks are formed when asymmetrical engulfment occurs, and how does the shape of droplet affect it?
4. In the last part of the manuscript, the author listed a variety of experimental protocols to change the molecular composition of their bilayer leaflets. Can the authors provide some

demonstrative experiments or relevant literature to prove the simulation results in this study?

This document contains our detailed point-by-point responses to all comments of the three reviewers. In our responses, we use blue text to refer to the document ‘Color-Marked-Manuscript.pdf’. In the latter document, our revisions are highlighted in blue, followed by a red bracket (Reviewer x , comment y) which refers to comment y of reviewer x .

The page and line numbers used below are identical to the page and line numbers of the document ‘Color-Marked-Manuscript.pdf’. Furthermore, *italic* style type is used for the text pieces quoted from the reports of the three reviewers.

Response to Reviewer 1

In the first paragraph of their report, Reviewer 1 says: *Ghosh and colleagues ... identify limits on the sizes of condensates that can be fully engulfed, showing that symmetric contraction of the membrane neck around the condensate is important for full engulfment and membrane fusion. ... Overall the work illustrates a potentially interesting mechanism.*

We thank Reviewer 1 for their positive overall assessment. In the following, we describe our point-by-point responses to the seven points raised by this reviewer:

1) *Very small vesicles (37 nm diameter) were used in the simulations. This represents a very high membrane curvature that is not very characteristic of the plasma membrane. Indeed, most endocytic structures (which have the opposite curvature) are larger than this. I can understand that very small vesicles were used because of the computational cost of simulating larger vesicles. The authors should acknowledge that the present case in non-physiological and comment on how the results might be different if the starting membrane were flatter. Would the same mechanisms be expected to be important?*

We use the term “endocytosis of liquid droplets” for any membrane process that starts with the engulfment of the droplet and then proceeds to the cleavage of the membrane neck. Furthermore, the high membrane curvature of the nanovesicles studied here applies to both synthetic liposomes, so-called small unilamellar vesicles (SUVs), as well as to extracellular vesicles such as exosomes and microvesicles which are produced by almost every living cell. In order to clarify this point, we have included another paragraph on small liposomes and extracellular vesicles in the revised manuscript, see the new blue lines 71 - 78 as well as the new references 36 - 42 which address both synthetic and cellular nanovesicles.

2) *How much the droplet spreads on the membrane will be controlled by the surface tension between the droplet and the surrounding solution, and the affinity between the droplet and the membrane surface. The authors seem to have fixed these parameters. How were these values chosen? With the chosen values, is the condensate more liquid-like or solid-like? If liquid-like, how would the viscosity of the condensate compare to literature values from experimental measurements?*

As explained in the *Methods*, our coarse-grained molecular dynamics approach (DPD) is based on different beads that represent different molecular groups and on force parameters between these beads as provided in Table 2. Using these parameters, we simulate a molecular system that consists of (i) a liquid mixture that undergoes liquid-liquid phase separation and (ii) lipid molecules that assemble into lipid bilayers in their fluid phase. The elastic moduli of these lipid bilayers are comparable to those measured experimentally. The phase diagram for the binary liquid mixture has been determined previously in our old Ref 59 = new Ref. 33, see new blue lines 40 - 41.

Because the droplets arise from liquid-liquid phase separation, these droplets are certainly liquid-like. We did not study the viscosity of the droplets but determined the interfacial tension $\Sigma_{\alpha\beta}$ of the liquid-liquid interface between the α droplet and the liquid bulk phase β . In general, the surface tension of condensate droplets can vary over several orders of magnitude as measured for aqueous two-phase systems, see our Ref 14. In order to clarify this point, we have now rephrased the corresponding text piece, see blue lines 22 - 26.

The nanodroplets studied here have a diameter between 11 and 19 nm. In order to obtain a well-defined and sharp interface for such small droplets, we chose a relatively large interfacial tension of about $3 k_B T/d^2$ or about 19 mN/m at room temperature which was included in Table 1 and which is now also mentioned in the added blue lines 143 - 145 and 148 - 151.

3) *Several groups have recently observed protein droplets to spread to a much greater extent on membrane surfaces, essentially creating a 2D patch on the membrane surface when they encounter it. (Snead and Gladfelter, NCB 2022; Yuan and Stachowiak, PNAS 2021, among several others.) How would that situation be different from what the authors present here? In particular, Yuan et al. showed that these 2D, patch-like condensates also generate inward membrane curvature. How might that mechanism be different (or similar) to the mechanism that the authors are investigating here? The authors should discuss this recent literature and explain how it relates to the present student.*

The extent of spreading depends on the contact angle θ_α^* of the α droplet. Depending on the intermolecular interactions, the contact angle can, in general, vary between $\theta_\alpha^* = 0^\circ$ for complete wetting by the α phase and $\theta_\alpha^* = 180^\circ$ for complete dewetting from the α phase. For our vesicle-droplet system, the contact angle is close to $\theta_\alpha^* = 90^\circ$, see Table S2. For complete wetting, the droplet forms a quasi-2D patch on the membrane as observed for some condensate droplets by Snead and Gladfelter, NCB 2022, as well as Yuan and Stachowiak, PNAS 2021. To clarify this issue, we have now added the new blue lines 52 - 58 where we also cite the two references mentioned by the reviewer as our new references 34 and 35.

4) *Along the same lines, a request for clarification: Why do the individual proteins that comprise the droplet not bind to the membrane surface? If the droplet can bind to the membrane, why can't the individual proteins bind? Will this happen if the protein-protein interactions are weakened in comparison to the protein-membrane interactions? Have the authors tried any simulations in that regime? This is an important question in the context of recent findings that membrane-binding may nucleate the assembly of condensates (Snead and Gladfelter, NCB 2022; Day and Stachowiak, NCB 2021).*

The binary liquid mixture considered here consists of two different types of beads, water beads and solute beads denoted by W and S. We explored solute concentrations that lead to liquid-liquid phase separation into solute-rich and water-poor α droplets immersed in a solute-poor and water-rich bulk phase β . The solute bead could represent a small macromolecule or a short peptide. The affinity of the water to the lipid head group and hydrocarbon chains is governed by the force parameter f_{HW} and f_{CW} , the affinity of the solute to the head group and hydrocarbon chains by f_{HS} and f_{CS} , respectively. In the present study, we reduced the number of parameters by choosing $f_{HW} = f_{HS}$ as well as $f_{CW} = f_{CS}$, corresponding to the same affinity of the W and S beads to the lipid head groups, see Table 2. To explain this point, we have now added the new blue lines 426 - 428 to the revised manuscript.

To describe preferred binding of the solute to the lipids, we would need to consider an affinity contrast between the interactions of the S and W beads with the lipid head groups. Such an affinity contrast has been explored in a previous study on planar bilayers, see our old Ref 38 = new Ref 51, and led to similar results as far as the closure of the membrane necks is concerned. This aspect has now been included into the revised manuscript, see the new blue lines 428 - 432 where we also cite the above-mentioned studies by Day and Stachowiak, NCB 2021, as well as by Snead and Gladfelter, NCB 2022, corresponding to our (old) Ref 27 and to our new Ref 35, respectively.

5) *I had a hard time understanding the idea of a "tight-lipped shape" to which the authors refer? How exactly is this defined geometrically? Can the authors quantify it in terms of geometrical parameters and report it for the different conditions that they simulated? As it is a major point in the study, I think it should be quantified.*

First, we want to emphasize that the simulation snapshots in Fig. 4 were intended to directly visualize the tight-lipped shape of the membrane necks. We agree, however, that the presentation in this figure was incomplete and somewhat confusing. In order to improve this presentation, we have now revised Fig. 4, emphasizing the different shapes of the contact lines by enframing them with white dashed circles and rectangles, see new bottom rows of Fig. 4a and b.

In addition, our terminology might have been somewhat confusing because we display half cuts rather than cross-sections. Thus, we have now replaced the term 'cross-sectional views' to 'half cut view' in the insets of Figs. 3 - 5.

The membrane necks are formed by the membrane segments adjacent to the contact lines. There-

fore, the strongly non-circular shapes of the contact lines imply strongly non-circular membrane necks. When the latter necks close, they attain a tight-lipped shape as can be visualized when we compare top views of the half cut shapes with front views and oblique side views. These different views are now displayed for both Fig. 4a and Fig. 4b in the new figures S1 and S2 in the Supplementary Information, see also new blue lines 225 - 227 in to the main text.

6) *It was not clear to me why a non-axisymmetric membrane neck corresponds to the "tight-lipped" shape. Can the authors explain?*

The membrane neck becomes non-axisymmetric when the line tension of the contact line becomes negative, see the data for the line tension in Fig. 7. To further elucidate this mechanism, we have now added the new blue lines 229 - 234 as well as a detailed explanation via the blue lines 309 - 319.

7) *How do the authors justify the observed fission/fusion? The driving force for it must come from the adhesion energy between the droplet and the membrane? Is this expected? Why doesn't the membrane simply coat the droplet, rather than undergoing full fission? Full fission implies that the two leaflets are being forced together with sufficient intensity to remove water molecules and suppress leaflet fluctuations. Would this be expected under physiological conditions or could it be an artifact of the authors' course-graining scheme?*

The adhesion energy together with the volume reduction provide the driving forces for the complete engulfment of the droplet. However, these driving forces act for all vesicle-droplet couples displayed in Fig. 3 and Fig. 4. The fission process, on the other hand, requires a relatively large stress asymmetry, as follows from a comparison of Fig. 2 with Fig. 3 and Fig. 4. This stress asymmetry plays the same role for nanovesicles as the spontaneous curvature for giant vesicles. In the latter case, the theory developed in the added Ref 61 predicts a curvature-induced constriction force that increases with the spontaneous curvature. When this curvature is sufficiently large, the constriction force acts to cleave the membrane neck as has been demonstrated experimentally for giant vesicles, new Ref 62, as we now explain by the new blue lines 341 - 348. Our study demonstrates an analogous fission mechanism for nanovesicles, with neck cleavage and vesicle division being controlled by the transbilayer stress asymmetry, see new blue lines 348 - 350.

Response to Reviewer 2

Reviewer 2 raises seven points which we will now address in a point-by-point manner:

1) *I found Fig 3-5 very qualitative and repetitive; they could have been summarized in a single figure. Also, it is unclear how relevant single snapshots are, and how statistically relevant the engulfment times are given that they have been reported for a single system realization. To be convincing, this data should be represented as a plot of quantitative measurements (e.g. times/membrane shapes vs lipid asymmetry) with proper statistical averages.*

As emphasized by the title of our manuscript, our study identifies and distinguishes different pathways for the endocytosis of droplets by nanovesicles. Figs. 3-5 illustrate these different pathways by providing examples for each pathway. Therefore, we strongly feel that these three figures are in no way repetitive. In addition, the different pathways can be distinguished by simply looking at the morphology of the vesicle-droplet systems which makes this distinction easily accessible to the interdisciplinary readership of Nature Comm. This advantage would be lost if we combined Figs. 3-5 into a single figure. Furthermore, our manuscript also provides a quantitative analysis of the different pathways, in terms of the stress asymmetries of the bilayers (Fig. 2), the mechanical tensions of the bilayer segments (Fig. 6) and the line tension of the contact line (Fig. 7).

2) *The volume control protocols used in the simulations need more clarification. It seems the authors control the vesicle volume during the simulation by varying the number of water molecules enclosed by it. However, the specific way in which this is implemented is not discussed. Also, which kind of a physiological situation this protocol of volume change would correspond to? Shouldn't that happen spontaneously in a simulation instead of being implemented by hand?*

Our protocol mimicks the process of osmotic deflation by which the vesicle volume is reduced via water permeation through the lipid bilayers. Water permeation is a relatively slow process. We

simulate this process by a sequence of volume reduction steps. In each step, the volume is reduced by moving subvolumes of water beads from the interior to the exterior solution, followed by a separate equilibration step. To further clarify this point, we have now added the blue lines 186 - 189, 222 - 223, and 454 - 460 to the revised manuscript.

3) *In more details, while the initial reduced volume values $\nu = \nu_0$ are explained in detail, the choice of volumes for the engulfment processes (for example $v = 0.6$) and the way this volume change is implemented (is it a sudden change or does it occur over a fixed time interval? If so, what is the deflation rate? How does this compare to deflation rates in experiments?) require an appropriate explanation. Looking at Ref. 39 it seems that in some cases deflation is sufficient to drive endocytosis of the exterior solution. If this is the case, what is the role of the droplet? Is the engulfment driven by the volume reduction or by the droplet or a combination of both? This whole aspect of the simulations needs to be explained in detail.*

The engulfment is driven by a combination of droplet adhesion and volume reduction as we now say explicitly in the added blue lines 186 - 189. Our protocol for volume reduction is now explained in detail, see new blue lines 454 - 460 and 222 - 223.

4) *The authors explore here the important role of bilayer asymmetry for these engulfment processes by artificially setting a specific asymmetry between the two leaflets which is then kept constant. However, lipid bilayers in nature display reshuffling such as flip-flop events. Even though such processes might not seem to be very rapid in cells, they are known to occur more frequently in coarse-grained molecular dynamics simulations. One wonders how they could affect the results presented here. Can the authors comment on this? Could flip-flops and similar reshuffling events change the asymmetry of the bilayer during the endocytosis and how would that affect the pathways? Have the bilayer been equilibrated before engulfment and the asymmetry measured after equilibration?*

We agree that lipid bilayers can contain molecular components such as cholesterol that undergo relatively fast flip-flops. In order to clarify this point, we have added the new blue lines 390 - 395 as well as the new Refs 64 - 67 to the revised manuscript.

5) *In their introduction the authors make a point of explaining the novelty of using deformable liquid droplets instead of rigid nanoparticles. However, the implications of this deformability for their results remains unclear.*

This comment is based on some misunderstanding. The main difference between liquid droplets and rigid (or solid) nanoparticles is not provided by their global deformability but by their local elastic properties. Rigid nanoparticles can be deformed to some extent by external forces and constraints but this deformability is always quite limited because rigid particles have a finite shear modulus which generates *local* restoring forces that act against the elastic deformations. To further emphasize this difference, we have added the new blue lines 82 - 85.

6) *Specifically, the deformability/fluidity of the droplet is not measured (is it even fluid? This should be measured.) or discussed enough to get an idea of how this setup compares to previous non-deformable nanoparticle engulfment works. Secondly, the deformation of the droplet during the endocytosis process (which seems visible in some snapshots and movies) is never properly characterized and its role is not discussed. Finally, the conclusions of the work seem to be quite independent of the droplet deformability and depend only on the droplet size. How different would these results be with a simple rigid nanoparticle of the same size that has been thoroughly explored before? This requires discussion and comparison to previous work.*

The droplets studied here are formed by liquid-liquid phase separation in the exterior solution and are, thus, necessarily fluid. In the revised manuscript, we now emphasize this point by the new blue lines 40 - 41 and explain it in more detail by rephrasing the paragraph on “Liquid-liquid phase separation in the exterior aqueous solution” in the *Methods* section, see new blue lines 498 - 504.

7) *Some minor notes:*

- *The movies would benefit from incorporating more frames to increase the time resolution.*

We have now added some text pieces to the movie captions in order to emphasize that the time-lapse movies consist of many frames that are only $0.5 \mu\text{s}$ apart. For $N_{ol} = 5500$, we added more frames to demonstrate that the morphology of two nested vesicles remains unchanged between $9 \mu\text{s}$

and 30 μs .

- Panel c) in Fig. 1 has a strange bit (like a half-erased bead) towards the top-right corner.

The figure has been amended by replacing the half bead by a full bead.

- Typo on Page 5: inequality for inequality.

This typo has been corrected.

Response to Reviewer 3

In the introductory paragraph, Reviewer 3 says: ... *the authors obtained two key parameters for endocytic pathways: transbilayer asymmetry of the vesicle membrane and the line tension of the membrane-droplet contact line. The vesicle-droplet systems used in the manuscript are concise and representative, which can reflect the physical relationship in some more complex systems such as the endocytic processes of biomolecular condensates.*

We thank Reviewer 3 for this rather positive overall assessment. Reviewer 3 also raised four points that we will now address in a point-by-point manner:

1. *In terms of interaction force field parameters, whether the interaction parameters of solute beads are adopted for the droplet? If not, what are the interaction parameters for droplets? And why the parameter for solute beads is set to be 70.*

The droplets studied here arise from liquid-liquid phase separation of a binary mixture consisting of water (W) and solute (S) beads. Thus, we distinguish three DPD force parameters as provided by f_{WW} between two water beads, f_{SS} between two solute beads, and f_{WS} between a water and a solute bead. The force parameter f_{WS} determines the interfacial tension $\Sigma_{\alpha\beta}$ of the liquid-liquid interface between the solute-rich droplet α and the solute-poor phase β . We choose the value $f_{\text{WS}} = 70$ to obtain a relatively large interfacial tension that ensures a sharp interface, see new blue lines 143 - 145 and 148 - 151.

2. *The diameters of the droplets used in the simulation are 14 d, 18.7 d, and 24.5 d, why did the authors choose the above size? For some droplets with smaller sizes (maybe 8 d, and 10 d), will the complete axisymmetric engulfment happen in the nanovesicles with $N_{\text{ol}} = 5700$ or 5963 lipids in the outer lipid bilayer?*

The size of the droplets is not controlled directly but arises via liquid-liquid phase separation for a certain prescribed solute concentration. To further clarify this point, we have now added the blue lines 42 - 46 and rephrased the paragraph of the *Methods* section on “Liquid-liquid phase separation in the exterior aqueous solution”, see the new blue lines 498 - 504, and

3. *The author should further analyze why the zip like necks are formed when asymmetrical engulfment occurs, and how does the shape of droplet affect it?*

The zip like necks, which we call tight-lipped necks, are formed for negative values of the line tension of the contact line. This mechanism is now emphasized in the new blue lines 229 - 234 and explained in more detail by the added blue lines 309-319.

4. *In the last part of the manuscript, the author listed a variety of experimental protocols to change the molecular composition of their bilayer leaflets. Can the authors provide some demonstrative experiments or relevant literature to prove the simulation results in this study?*

In the last paragraph of the *Summary and Outlook* section, we propose an experimental procedure that combines changes in the molecular composition of the bilayer membranes with osmotic deflation of the nanovesicle. As discussed in this paragraph, a variety of protocols have been developed by which one can change the molecular composition of the bilayers. However, we are not aware of an experimental study in which such a compositional change was combined with osmotic deflation. Therefore, such a combination remains a challenge for future experiments.